# Distinct inactive conformations of the dopamine D2 and D3 receptors correspond to different extents of inverse agonism

**J Robert Lane[1,2]\*, Ara M Abramyan[3], Pramisha Adhikari[3], Alastair C Keen[1,2,4], Kuo-Hao Lee[3], Julie Sanchez[1,2], Ravi Kumar Verma[3], Herman D Lim[4], Hideaki Yano[3], Jonathan A Javitch[5,6,7]\*, Lei Shi[3]\***

[1]Division of Pharmacology, Physiology and Neuroscience, School of Life Sciences, Queen's Medical Centre, University of Nottingham, Nottingham, United Kingdom; [2]Centre of Membrane Protein and Receptors, Universities of Birmingham and Nottingham, Nottingham, United Kingdom; [3]Computational Chemistry and Molecular Biophysics Unit, National Institute on Drug Abuse - Intramural Research Program, National Institutes of Health, Baltimore, United States; [4]Drug Discovery Biology, Department of Pharmacology and Medicinal Chemistry, Monash Institute of Pharmaceutical Sciences, Monash University, Parkville, Australia; [5]Department of Psychiatry, Vagelos College of Physicians and Surgeons, Columbia University, New York, United States; [6]Division of Molecular Therapeutics, New York State Psychiatric Institute, New York, United States; [7]Department of Pharmacology, Vagelos College of Physicians and Surgeons, Columbia University, New York, United States

**\*For correspondence:**
Rob.Lane@nottingham.ac.uk (JRL);
jaj2@cumc.columbia.edu (JAJ);
lei.shi2@nih.gov (LS)

**Competing interests:** The authors declare that no competing interests exist.

**Abstract** By analyzing and simulating inactive conformations of the highly homologous dopamine $D_2$ and $D_3$ receptors ($D_2R$ and $D_3R$), we find that eticlopride binds $D_2R$ in a pose very similar to that in the $D_3R$/eticlopride structure but incompatible with the $D_2R$/risperidone structure. In addition, risperidone occupies a sub-pocket near the $Na^+$ binding site, whereas eticlopride does not. Based on these findings and our experimental results, we propose that the divergent receptor conformations stabilized by $Na^+$-sensitive eticlopride and $Na^+$-insensitive risperidone correspond to different degrees of inverse agonism. Moreover, our simulations reveal that the extracellular loops are highly dynamic, with spontaneous transitions of extracellular loop 2 from the helical conformation in the $D_2R$/risperidone structure to an extended conformation similar to that in the $D_3R$/eticlopride structure. Our results reveal previously unappreciated diversity and dynamics in the inactive conformations of $D_2R$. These findings are critical for rational drug discovery, as limiting a virtual screen to a single conformation will miss relevant ligands.

## Introduction

G-protein-coupled receptors (GPCRs) are important therapeutic targets for numerous human diseases. Our understanding of GPCR functional mechanisms has evolved from a simple demarcation of single active and inactive states to the appreciation and detection of multiple active states responsible for partial or biased agonism (*Latorraca et al., 2017*; *Venkatakrishnan et al., 2013*; *Weis and Kobilka, 2018*). High-resolution crystal structures of these proteins are vital for structure-based (rational) drug discovery (RDD) efforts designed to tailor selectivity and efficacy (*Congreve et al., 2014*; *Michino et al., 2015a*). While considerable efforts have been directed at

**eLife digest** Almost a third of prescribed drugs work by acting on a group of proteins known as GPCRs (short for G-protein coupled receptors), which help to transmit messages across the cell's outer barrier. The neurotransmitter dopamine, for instance, can act in the brain and body by attaching to dopamine receptors, a sub-family of GPCRs. The binding process changes the three-dimensional structure (or conformation) of the receptor from an inactive to active state, triggering a series of molecular events in the cell.

However, GPCRs do not have a single 'on' or 'off' state; they can adopt different active shapes depending on the activating molecule they bind to, and this influences the type of molecular cascade that will take place in the cell. Some evidence also shows that classes of GPCRs can have different inactive structures; whether this is also the case for the dopamine $D_2$ and $D_3$ receptors remained unclear. Mapping out inactive conformations of receptors is important for drug discovery, as compounds called antagonists can bind to inactive receptors and interfere with their activation.

Lane et al. proposed that different types of antagonists could prefer specific types of inactive conformations of the dopamine $D_2$ and $D_3$ receptors. Based on the structures of these two receptors, the conformations of $D_2$ bound with the drugs risperidone and eticlopride (two dopamine antagonists) were simulated and compared. The results show that the inactive conformations of $D_2$ were very different when it was bound to eticlopride as opposed to risperidone. In addition $D_2$ and $D_3$ showed a very similar conformation when attached to eticlopride. The two drugs also bound to the inactive receptors in overlapping but different locations. These computational findings, together with experimental validations, suggest that $D_2$ and $D_3$ exist in several inactive states that only allow the binding of specific drugs; these states could also reflect different degrees of inactivation. Overall, the work by Lane et al. contributes to a more refined understanding of the complex conformations of GPCRs, which could be helpful to screen and develop better drugs.

---

the development of biased agonists that couple preferentially to a particular effector pathway (*Free et al., 2014*; *Manglik et al., 2016*; *McCorvy et al., 2018*), less attention has been dedicated to the possibility that different antagonist scaffolds with differing efficacy of inverse agonism might lead to different receptor conformations and hence different 'inactive' states. Such a possibility could have a major impact on RDD for antagonists, since a GPCR crystal structure stabilized by a particular antagonist might represent an invalid docking target for an antagonist that prefers a different inactive conformation. Although substantial differences in antagonist binding mode and position of the binding pockets have been revealed among different aminergic receptors, no conformational differences has been detected for the inactive state in any individual aminergic receptor (*Michino et al., 2015a*). In particular, although a number of antagonists derived from different scaffolds have been co-crystallized with the $\beta_2$ adrenergic receptor, conformational differences among these crystal structures are minimal (*Michino et al., 2015a*).

Curiously, the inactive state structures of the highly homologous dopamine D2 and D3 receptors ($D_2R$ and $D_3R$) revealed substantial differences on the extracellular side of the transmembrane domain, especially in TM6 (*Figure 1*), when bound with antagonists derived from different scaffolds (*Chien et al., 2010*; *Wang et al., 2018*). Specifically, the $D_3R$ structure is in complex with eticlopride, a substituted benzamide (PDB: 3PBL) (*Chien et al., 2010*), while the $D_2R$ structure is bound with risperidone, a benzisoxazole derivative (PDB: 6CM4) (*Wang et al., 2018*). The binding poses of the two ligands differ substantially. Risperidone is oriented relatively perpendicular to the membrane plane with its benzisoxazole ring penetrating into a hydrophobic pocket beneath the orthosteric binding site (OBS) of $D_2R$; in contrast, eticlopride is oriented relatively parallel to the membrane plane and contacts the extracellular portion of TM5 in $D_3R$, a sub-pocket that risperidone does not occupy in $D_2R$ (*Sibley and Shi, 2018*; *Wang et al., 2018*). Nemonapride, another substituted benzamide, binds in the OBS of the slightly divergent $D_4R$ (PDB: 5WIV) (*Wang et al., 2017*) in a manner very similar to that of eticlopride in the $D_3R$ (*Sibley and Shi, 2018*).

Importantly, the co-crystalized ligands (risperidone, eticlopride, and nemonapride) display little subtype selectivity across $D_2R$, $D_3R$, and $D_4R$ (*Chien et al., 2010*; *Hirose and Kikuchi, 2005*; *Silvestre and Prous, 2005*; *Wang et al., 2017*) (also see PDSP database; *Roth et al., 2000*). Given

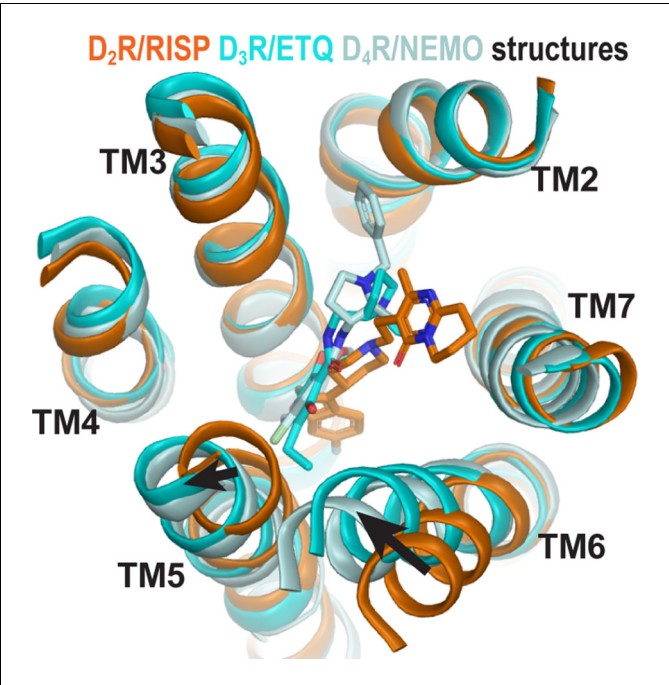

**Figure 1.** The structures of homologous $D_2R$, $D_3R$, and $D_4R$ show different conformations in the extracellular vestibules. Superpositioning of $D_2R$, $D_3R$, and $D_4R$ structures shows that the binding of eticlopride (ETQ, cyan) in $D_3R$ and nemonapride (NEMO, pale cyan) in $D_4R$ result in outward and inward rearrangements of the extracellular portions of TM5 and TM6, respectively, compared to the binding of risperidone (RISP, orange) in $D_2R$.

The online version of this article includes the following figure supplement(s) for figure 1:

**Figure supplement 1.** Chemical structure alignments of the non-selective $D_2$-like receptors ligands.

---

the high homology among these $D_2$-like receptors, especially between $D_2R$ and $D_3R$, the drastic conformational differences between the inactive state structures of these receptors may be better explained by different binding poses of antagonists bearing different scaffolds rather than inherent differences in the receptors. Thus, we hypothesized that different antagonist scaffolds may favor distinct inactive conformations of $D_2R$. To test this hypothesis, we carried out extensive molecular dynamics (MD) simulations of $D_2R$ in complex with non-selective antagonists derived from different scaffolds to characterize the plasticity of the OBS and the extracellular loop dynamics in the inactive conformational state.

## Results

### The Ile$^{3.40}$ sub-pocket is occupied by risperidone and spiperone but not eticlopride in $D_2R$

Compared to eticlopride bound in the $D_3R$ structure, risperidone in the $D_2R$ structure penetrates deeper into the binding site, with its benzisoxazole moiety occupying a sub-pocket that eticlopride does not reach. By examining the $D_2R$/risperidone structure, we found that the benzisoxazole moiety is enclosed by eight residues in $D_2R$, which are identical among all $D_2$-like receptors (i.e. $D_2R$, $D_3R$, and $D_4R$): Cys118$^{3.36}$ (superscripts denote Ballesteros-Weinstein numbering *Ballesteros and Weinstein, 1995*), Thr119$^{3.37}$, Ile122$^{3.40}$, Ser197$^{5.46}$, Phe198$^{5.47}$, Phe382$^{6.44}$, Trp386$^{6.48}$, and Phe390$^{6.52}$. Notably, three of these residues (Ile122$^{3.40}$, Phe198$^{5.47}$, and Phe382$^{6.44}$) on the intracellular side of the OBS that we previously defined (*Michino et al., 2015a*), accommodate the F-substitution at the tip of the benzisoxazole ring in a small cavity (termed herein as the Ile$^{3.40}$ sub-pocket) (*Figure 2a*). Both Ile122$^{3.40}$ and Phe382$^{6.44}$ of this Ile$^{3.40}$ sub-pocket are part of the conserved Pro$^{5.50}$-Ile$^{3.40}$-Phe$^{6.44}$ motif that undergoes rearrangement upon receptor activation (*Rasmussen et al., 2011*), and

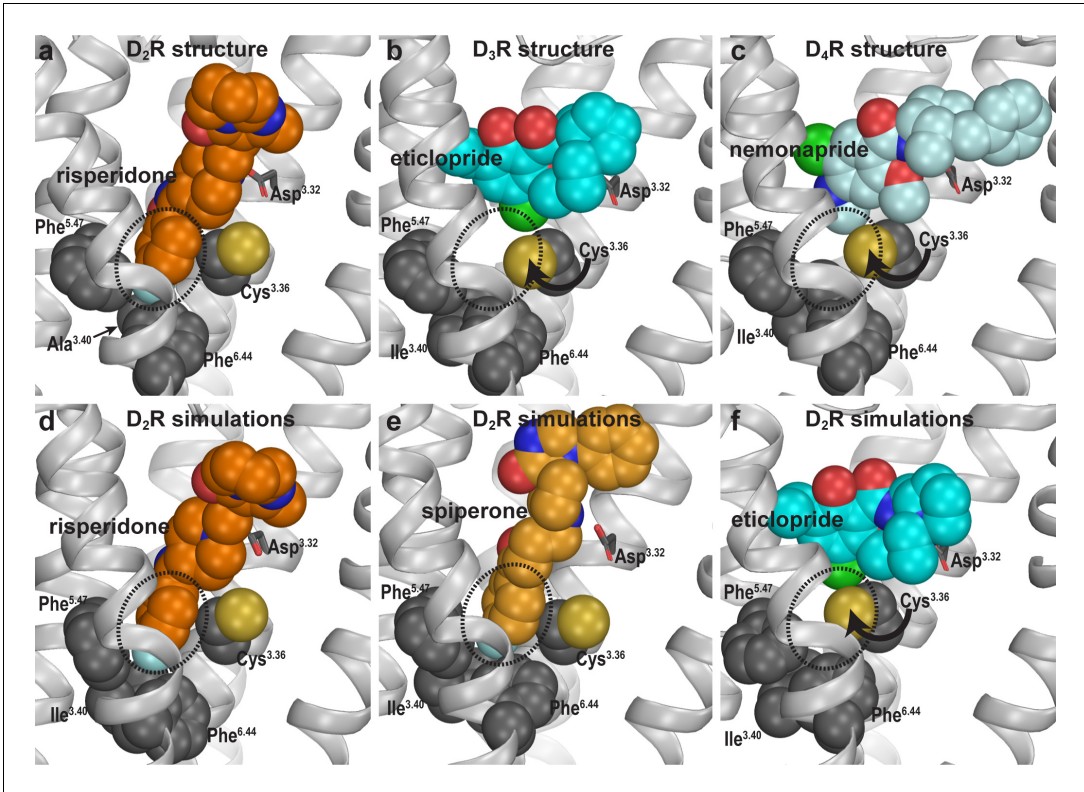

**Figure 2.** Divergent occupations of the Ile[3.40] sub-pocket by non-selective ligands from different scaffolds. In the D$_2$R structure (**a**), the F-substitution on the benzisoxazole ring of risperidone occupies the Ile[3.40] sub-pocket (dotted circle) enclosed by conserved Ile[3.40] (mutated to Ala in the crystal structure to thermostabilize the receptor), Phe[5.47], and Phe[6.44]. The same viewing angle shows that in the D$_3$R (**b**) and D$_4$R (**c**) structures, Cys[3.36] rotates to fill in the Ile[3.40] sub-pocket, and the substituted benzamides eticlopride and nemonapride cannot occupy the aligned sub-pockets. In our D$_2$R/risperidone simulations (**d**), risperidone maintains its pose revealed by the crystal structure. In the D$_2$R/spiperone simulations (**e**), the Ile[3.40] sub-pocket is similarly occupied as in D$_2$R/risperidone. In the D$_2$R/eticlopride simulations (**f**), the Ile[3.40] sub-pocket is collapsed as in the D$_3$R (**b**) and D$_4$R (**c**) structures (this trend is independent of the force field being used in the simulations).
The online version of this article includes the following figure supplement(s) for figure 2:

**Figure supplement 1.** Allosteric communication between the Ile[3.40] sub-pocket and the Na$^+$ binding site.

we have found that the I122[3.40]A mutation renders D$_2$R non-functional (**Klein Herenbrink et al., 2019**; **Wang et al., 2018**). Interestingly, this Ile[3.40] sub-pocket is collapsed in both the D$_3$R and D$_4$R structures (**Sibley and Shi, 2018**; **Figure 2b,c**). We noted that this collapse is associated with rotation of the sidechain of Cys[3.36]: In the D$_2$R/risperidone structure, the sidechain of Cys[3.36] faces the OBS, whereas in the D$_3$R/eticlopride and D$_4$R/nemonapride structures, it rotates downwards to partially fill the Ile[3.40] sub-pocket (**Figure 2a–c**).

To test our hypothesis that these observed differences in the crystal structures are due to the binding of antagonists bearing different scaffolds but not intrinsic divergence of D$_2$-like receptors, we compared the binding modes of three non-selective antagonists in D$_2$R. We reverted three thermostabilizing mutations introduced for crystallography (I122[3.40]A, L375[6.37]A, and L379[6.41]A) back to their WT residues, established WT D$_2$R models in complex with risperidone, spiperone, or eticlopride, and carried out extensive MD simulations (see Materials and methods, **Figure 1—figure supplement 1** and **Table 1**).

In our prolonged MD simulations of the WT D$_2$R/risperidone complex (>65 μs, **Table 1**), we observed that risperidone stably maintains the binding pose captured in the crystal structure, even without the thermostabilizing mutations (**Figure 2d**). Thus, the I122[3.40]A mutation has minimal impact on the binding pose of risperidone. Interestingly, in the simulations of the WT D$_2$R model in

**Table 1.** Summary of molecular dynamics simulations.

| Receptor | Ligand | Bound na$^+$ | Number of OPLS3e trajectories | Number of CHARMM36 trajectories | Accumulated simulation time (ns) |
|---|---|---|---|---|---|
| D$_2$R | Risperidone | + | 12 | | 28410 |
| | | - | 11 | | 42240 |
| | Spiperone | + | 22 | | 42000 |
| | | - | 17 | | 29550 |
| | Eticlopride | + | 5 | 12 | 51540 |
| | | - | 7 | | 11280 |
| | (-)-Sulpiride | + | 3 | | 4500 |
| | | - | 3 | | 3600 |
| | Aripiprazole | + | 40 | | 66660 |
| D$_3$R | Eticlopride | + | | 3 | 13200 |
| | | - | | 4 | 6240 |
| | R22 | + | | 7 | 33600 |
| | S22 | - | | 7 | 59400 |
| Total | | | 120 | 33 | 392220 |

complex with spiperone, a butyrophenone derivative, the F-substitution on the butyrophenone ring similarly occupies the Ile$^{3.40}$ sub-pocket as risperidone (*Figure 2e*). Note that the F-substitutions in risperidone and spiperone are located at similar distances to the protonated N atoms that interact with Asp$^{3.32}$ (measured by the number of carbon atoms between them, *Figure 1—figure supplement 1*) and these two ligands appear to be optimized to occupy the Ile$^{3.40}$ sub-pocket.

In contrast, in our simulations of the D$_2$R/eticlopride complex, the eticlopride pose revealed in the D$_3$R structure (PDB: 3PBL) is stable throughout the simulations and does not protrude into the Ile$^{3.40}$ sub-pocket (*Figure 2f*). Consistent with the difference in the crystal structures noted above (*Figure 2a,b*), when risperidone and spiperone occupy the Ile$^{3.40}$ sub-pocket, the sidechain of Cys118$^{3.36}$ rotates away with its χ1 rotamer in *gauche-*, while in the presence of the bound eticlopride, this rotamer is stable in *trans* (*Figure 2—figure supplement 1*).

To validate these computational findings regarding the occupation of the Ile$^{3.40}$ sub-pocket, we mutated Ile122$^{3.40}$ of WT D$_2$R to both Trp and Ala and characterized how these mutations affect the binding affinities for spiperone, risperidone, and eticlopride (*Table 2*). We hypothesized that the bulkier sidechain of Trp at position 3.40 would hamper the binding of spiperone and risperidone

**Table 2.** The effect of mutations on the binding affinities of selected D$_2$R ligands.
The affinities of [3H]spiperone were determined in saturation experiments at WT or mutant SNAP-tagged D$_{2S}$Rs stably expressed in FlpIn CHO cells. Binding affinity values for risperidone and eticlopride were obtained in competition binding experiments. Means of n independent experiments performed in triplicate are shown with 95% confidence intervals.

| SNAP-D$_{2S}$R | [3H]spiperone saturation binding p$K_d$ ($K_d$, nM) (95% CI) | N | [3H]spiperone competition binding Risperidone p$K_i$ ($K_i$, nM) (95% CI) | N | Eticlopride p$K_i$ ($K_i$, nM) (95% CI) | N |
|---|---|---|---|---|---|---|
| WT | 9.74 (0.18) (9.36–10.14) | 3 | 8.55 (2.8) (8.07–9.04) | 8 | 9.84 (0.14) (9.10–10.58) | 3 |
| WT -Na$^+$ | 9.70 (0.20) (9.09–10.32) | 3 | 8.96 (1.1) (8.84–9.08) | 6 | - | |
| I122$^{3.40}$A | 9.74 (0.18) (9.09–10.38) | 3 | 8.14 (7.9) (7.97–8.32) | 8 | 10.33 (0.04) (10.22–10.44) | 3 |
| I122$^{3.40}$W | 8.95 (1.15) (8.59–9.30) | 3 | 7.43 (37) (7.11–7.75) | 5 | 9.61 (0.25) (9.33–9.89) | 4 |

since they occupy the Ile$^{3.40}$ sub-pocket but have no effect on eticlopride binding, while the smaller Ala should not affect the binding of spiperone or risperidone. Consistent with this hypothesis, the I122W mutation decreased the binding affinities of risperidone (13-fold) and spiperone (6-fold) compared to WT but had no effect on that of eticlopride. In contrast, the I122A mutation did not affect the affinities of spiperone or risperidone, which is consistent with our simulation results that show the I122A mutation has minimal impact on risperidone binding. In contrast, I122A caused a threefold increase in the affinity of eticlopride, suggesting that the I122A mutation may promote an inactive conformation of D$_2$R that favors eticlopride binding. Together these results support our proposal that different antagonist scaffolds may favor distinct inactive conformations of D$_2$R.

## Occupation of the Ile$^{3.40}$ sub-pocket confers insensitivity to Na$^+$ in antagonist binding

Ligand binding in D$_2$-like receptors can be modulated by Na$^+$ bound in a conserved allosteric binding pocket coordinated by Asp$^{2.50}$ and Ser$^{3.39}$ (*Michino et al., 2015b*; *Neve, 1991*; *Wang et al., 2017*). Note that the aforementioned Cys$^{3.36}$ and Ile$^{3.40}$ are adjacent to the Na$^+$ coordinating Ser$^{3.39}$; thus, we further hypothesized that the occupation of the Ile$^{3.40}$ sub-pocket by spiperone or risperidone makes them insensitive to Na$^+$. To test this hypothesis, we simulated D$_2$R/risperidone, D$_2$R/spiperone, D$_2$R/eticlopride, and D$_2$R/(-)-sulpiride complexes in the presence versus absence of bound Na$^+$ (*Table 1*). Interestingly, the occupancy of the Ile$^{3.40}$ sub-pocket by either spiperone or risperidone was unaffected by the presence or absence of bound Na$^+$ (*Figure 2—figure supplement 1*). In contrast, while the poses of eticlopride and (-)-sulpiride are highly stable in the presence of bound Na$^+$, they oscillated between different poses in the absence of Na$^+$. These oscillations are associated with the sidechain of Cys$^{3.36}$ swinging back and forth between the two rotamers, suggesting an important role of Na$^+$ binding in stabilizing the poses of eticlopride and (-)-sulpiride and the configuration of the Ile$^{3.40}$ sub-pocket (*Figure 2—figure supplement 1*). Interestingly, the previous MD simulations described by Wang et al. indicated that nemonapride's binding pose in D$_4$R is more stable in the presence of bound Na$^+$ as well (*Wang et al., 2017*).

Consistent with these computational results, we have previously shown that spiperone binding is insensitive to the presence of Na$^+$, while the affinities of eticlopride and sulpiride are increased in the presence of Na$^+$ (*Michino et al., 2015b*). In this study, we performed binding experiments in the absence or presence of Na$^+$ and found the affinity of risperidone to be unaffected, in accordance with this hypothesis (*Table 2*).

Together these findings support our hypothesis that the ability of a ligand to bind the Ile$^{3.40}$ sub-pocket relates with its sensitivity to Na$^+$ in binding, due to allosteric connections between the sub-pocket and the Na$^+$ binding site.

## Functional consequences of distinct antagonist-bound inactive conformations

To further investigate the functional impact of these conformational differences surrounding the OBS, we used a bioluminescence resonance energy transfer (BRET) assay, which measures conformational changes of the Go protein heterotrimer following activation by D$_2$R (*Michino et al., 2017*), to evaluate the inverse agonism activities of several representative D$_2$R ligands. These ligands can be categorized into two groups according to their sensitivities to Na$^+$ in binding at D$_2$R, which have been characterized either in our current study or in previous studies (*Michino et al., 2015b*; *Neve, 1991*; *Newton et al., 2016*). While risperidone, spiperone, and (+)-butaclamol have been found to be insensitive to Na$^+$ in binding, (-)-sulpiride, eticlopride, and raclopride show enhanced binding affinities in the presence of Na$^+$. Using quinpirole as a reference full agonist, we found that the Na$^+$ insensitive ligands display significantly greater inverse agonism ($< -30\%$ that of the maximal response of quinpirole) relative to the Na$^+$-sensitive ligands ($> -15\%$ that of the maximal response of quinpirole, *Figure 3*). These observations are consistent with findings from earlier [$^{35}$S]GTP$\gamma$S binding experiments of Roberts and Strange in which (+)-butaclamol, risperidone, and spiperone were found to inhibit significantly more [$^{35}$S]GTP$\gamma$S binding than raclopride and (-)-sulpiride (*Roberts and Strange, 2005*). Of note, these [$^{35}$S]GTP$\gamma$S-binding experiments were performed in the absence of Na$^+$.

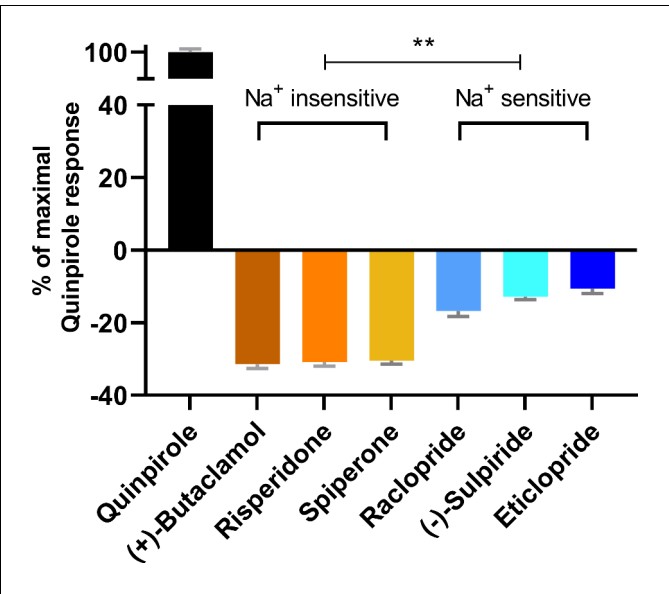

**Figure 3.** The extent of inverse agonism is negatively related with the $Na^+$ sensitivity of ligand binding. In a $D_2R$-Go BRET assay, the maximal responses of the indicated ligands are normalized to that of the reference full agonist quinpirole. The ligands that are insensitive to $Na^+$ in $D_2R$ binding display significantly higher inverse agonism (in each case, **$p < 0.0001$ using ordinary one-way ANOVA followed by Tukey's multiple comparisons test) than the $Na^+$-sensitive ligands; however, within the $Na^+$-sensitive group, raclopride is significantly different from eticlopride ($p = 0.005$).

Based on these functional data together with the different binding modes revealed by our computational simulations, we propose that ligands that occupy the $Ile^{3.40}$ sub-pocket exhibit a greater level of inverse agonism as compared to those that do not. Therefore, across the tested inverse agonists there is a negative relation between ligand sensitivity to $Na^+$ and the extent of inverse agonism at $D_2R$. The differential occupation of the $Ile^{3.40}$ sub-pocket is the structural basis for the $Na^+$ sensitivity, which contributes significantly to the extent of inverse agonism of the tested ligands.

## Plasticity of the ligand-binding site propagates to affect the overall receptor conformation

By occupying the $Ile^{3.40}$ sub-pocket, the benzisoxazole moiety of risperidone pushes the conserved $Phe^{6.52}$ away from the binding site in the $D_2R$/risperidone structure compared to its position in the $D_3R$/eticlopride structure. This interaction is responsible for positioning the aromatic cluster of TM6 and TM7 ($Trp^{6.48}$, $Phe^{6.51}$, $Phe^{6.52}$, $His^{6.55}$, and $Tyr^{7.35}$) in $D_2R$ differently from its configurations in the $D_3R$ and $D_4R$ structures, resulting in an overall outward positioning of the extracellular portion of TM6 in $D_2R$ (*Figure 4—figure supplement 1*). On the extracellular side of the OBS, the space near $Ser^{5.42}$ and $Ser^{5.43}$ that accommodates the bulky substitutions of the benzamide rings of the bound eticlopride and nemonapride in the $D_3R$ and $D_4R$ structures is not occupied by risperidone in $D_2R$, which is likely associated with the inward movement of the extracellular portion of TM5 in $D_2R$ relative to those in the $D_3R$ and $D_4R$ structures (*Figure 1*).

To evaluate whether these conformational rearrangements are due to the minor divergence in these regions of the receptors or to the ligand-binding site plasticity that accommodates ligands bearing different scaffolds, we compared the resulting conformations of $D_2R$ bound with risperidone or eticlopride. We observed the same trend of rearrangements of the transmembrane segments surrounding the OBS in the resulting receptor conformations from our $D_2R$/risperidone and $D_2R$/eticlopride simulations (*Figure 4a*), that is, an inward movement of TM6 and outward movement of TM5 in the presence of the bound eticlopride (*Figure 4b,c*). Without such movements in $D_2R$/eticlopride, $Ser193^{5.42}$ and $Ser194^{5.43}$ would clash with the bound eticlopride (*Figure 4a*). These

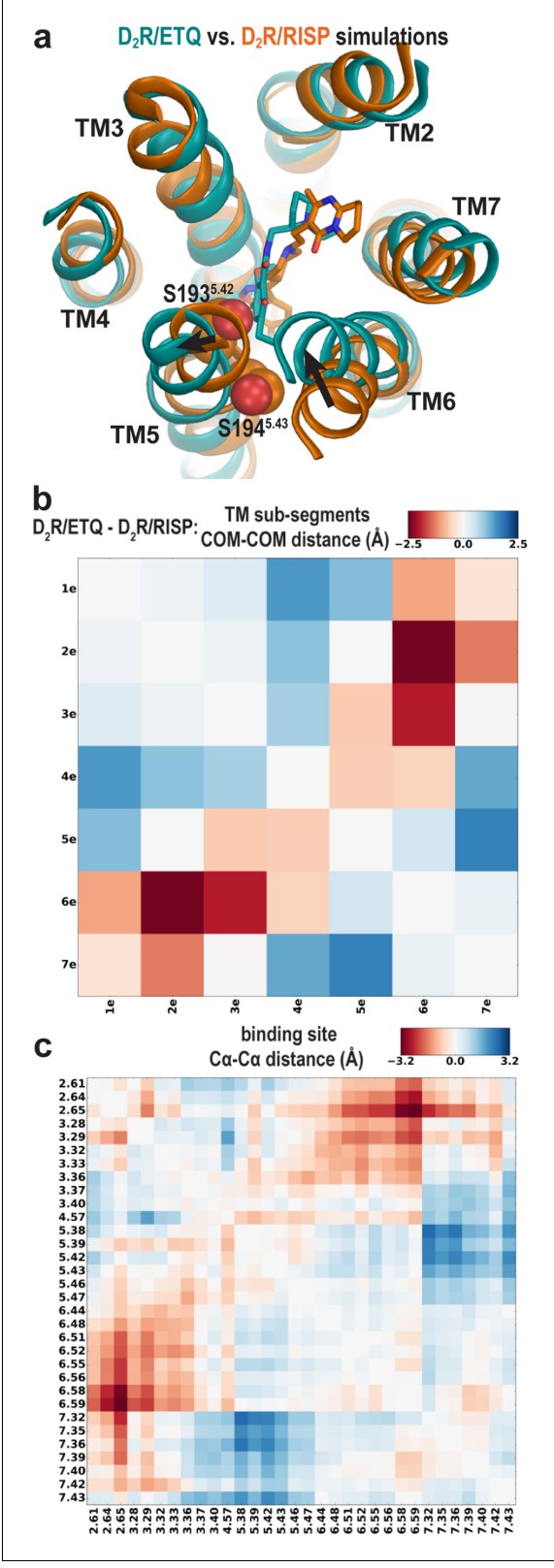

**Figure 4.** The different conformations in the extracellular vestibules of $D_2R$ and $D_3R$ are likely due to binding of non-selective ligands from different scaffolds. (a) Compared to the comparison of the crystal structures shown in *Figure 1*, superpositioning of representative frames of the $D_2R/ETQ$ and $D_2R/RISP$ simulations shows a similarly trend of the outward and inward movements of TM5 and TM6, respectively, in the presence of the bound ETQ, *Figure 4 continued on next page*

*Figure 4 continued*

even when the simulations were started from the D$_2$R conformation stabilized by RISP. Note Ser193$^{5.42}$ and Ser194$^{5.43}$ would clash with the bound eticlopride if there was no conformational adjustment. (b, c) PIA-GPCR analysis (see Materials and methods) comparing the D$_2$R/ETQ and D$_2$R/RISP conformations. The analysis of the pairwise-distance differences among the subsegments (b) indicates that TM6e moves inward (smaller distance to TM2e, dark red pixel), while TM5e moves outward (larger distances to TM7e, dark blue pixel) in the D$_2$R/ETQ simulations. The analysis of pairwise-distance differences among the Cα atoms of the ligand-binding residues (c) indicates significant changes near residues Phe189$^{5.38}$, Ser193$^{5.42}$, Asn367$^{6.58}$, and Ile368$^{6.59}$ (darker colored pixels). The online version of this article includes the following figure supplement(s) for figure 4:

**Figure supplement 1.** The occupation of the Ile$^{3.40}$ pocket by risperidone is associated with outward movement of the extracellular portion of TM6.

findings further support our inference that differences between the D$_2$R and D$_3$R inactive structures are largely due to the different scaffolds of the bound non-selective ligands.

## The extracellular loop 2 (EL2) of D$_2$R/risperidone can spontaneously unwind

In addition to differences in the transmembrane segments surrounding the OBS, there are also substantial differences in the configuration of EL2 in the D$_2$R and D$_3$R structures. EL2 between TM4 and TM5 is connected to TM3 via a disulfide bond formed between Cys$^{EL2.50}$ (see Materials and methods and *Figure 5—figure supplement 1* for the indices of EL1 and EL2 residues) and Cys$^{3.25}$. The conformation of EL2, the sequence of which is not conserved among aminergic GPCRs, is expected to be dynamic. Indeed, in the D$_2$R/risperidone structure, the sidechains of residues 176$^{EL2.40}$, 178$^{EL2.46}$, 179$^{EL2.47}$, and 180$^{EL2.48}$, which are distal to the OBS were not solved, likely due to their dynamic nature. Curiously, the portion of EL2 C-terminal to Cys182$^{EL2.50}$ (residues 182$^{EL2.50}$-186$^{EL2.54}$), which forms the upper portion of the OBS that is in contact with ligand, is in a helical conformation in the D$_2$R/risperidone structure.

Strikingly, in our MD simulations of D$_2$R complexes, we found that this helical region showed a tendency to unwind (*Video 1*). The unwinding of EL2 involves a drastic rearrangement of the sidechain of Ile183$^{EL2.51}$, which dissociates from a hydrophobic pocket formed by the sidechains of Val111$^{3.29}$, Leu170$^{4.60}$, Leu174$^{EL2.38}$, and Phe189$^{5.38}$. Specifically, the unwinding process is initiated by the loss of a hydrogen-bond (H-bond) interaction between the sidechain of Asp108$^{3.26}$ and the backbone amine group of Ile183$^{EL2.51}$ formed in the D$_2$R/risperidone structure (*Figure 5—figure supplement 2b*, step (i). When this interaction is broken, the orientation of residues 182$^{EL2.50}$-186$^{EL2.54}$ deviates markedly from that of the crystal structure, losing its helical conformation (see below). Subsequently, the sidechain of Ile183$^{EL2.51}$ rotates outwards and passes a small steric barrier of Gly173$^{EL2.37}$ (*Figure 5—figure*

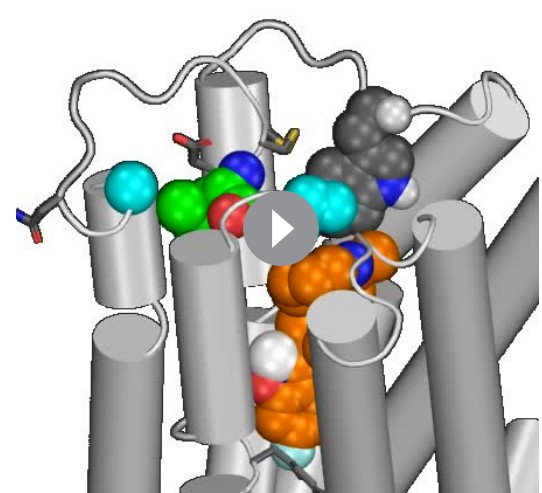

**Video 1.** A movie of a 4.2 μs D$_2$R/risperidone trajectory collected using the OPLS3e force field shows spontaneous unwinding of EL2. The conformation of EL2 gradually transitions to an extended configuration similar to that in the D3R structure. See *Figure 5—figure supplement 2* for the pathway of unwinding. Note that the extended conformation of EL2 stabilizes Trp100$^{EL1.50}$. The Cα atom of Gly173$^{EL2.37}$, the sidechains of Trp100$^{EL1.50}$, Ile183$^{EL2.51}$, and Ile184$^{EL2.52}$ and the bound risperidone are shown as spheres. Asp108$^{3.26}$ and the disulfide bond between Cys107$^{3.25}$ and Cys182$^{EL2.50}$ are shown as sticks. The carbon atoms of Gly173$^{EL2.37}$ and Ile184$^{EL2.52}$ are colored in cyan, those of Ile183$^{EL2.51}$ are in green, those of Trp100$^{EL1.50}$, Cys107$^{3.25}$, Asp108$^{3.26}$, Asn175$^{EL2.39}$, and Cys182 $^{EL2.50}$ are in dark gray; those of the bound ligand risperidone are in orange.

https://elifesciences.org/articles/52189#video1

*supplement 2b*, step (ii), and in some trajectories makes a favorable hydrophobic interaction with the sidechain of Ala177[EL2.45]. In a few long trajectories, Ile183[EL2.51] rotates further toward the extracellular vestibule where it can make favorable interactions with hydrophobic or aromatic residues from the N terminus, or the bound risperidone (*Video 1*). Consequently, residues 182[EL2.50]-186[EL2.54] are in a fully extended loop conformation while Ile184[EL2.52] tilts under EL2 (*Figure 5—figure supplement 2b*, step (iii).

In the $D_3R$ structure, the aligned residue for Asp108[3.26] of $D_2R$ is conserved as Asp104[3.26]; its sidechain forms an interaction not with Ile182[EL2.51] but rather with the sidechain of Asn173[EL2.39], which is also conserved in $D_2R$ as Asn175[EL2.39]. In the $D_4R$, the aligned two residues (Asp109[3.26] and Asn175[EL2.39]) are conserved as well, their sidechains are only 4.3 Å away in the $D_4R$ structure, a distance slightly larger than the 3.2 Å in the $D_3R$ structure. Even though these residues are conserved in $D_2R$, the interaction in $D_3R$ (and potentially in $D_4R$), between Asp[3.26]-Asn[EL2.39], is not present in the $D_2R$ structure in which the aligned Asn175[EL2.39] faces lipid (*Figure 5—figure supplement 2a*). However, in a few of our long $D_2R$ simulations, Asn175[EL2.39] gradually moves inwards and approaches Asp108[3.26] (*Figure 5—figure supplement 2b*, step (iv). At this point, the EL2 conformation of $D_2R$ is highly similar to that of $D_3R$ (*Figure 5—figure supplement 2c*), suggesting that EL2 is dynamic and can exist in both conformations.

We evaluated the tendency of the EL2 helix to unwind in each of the simulated $D_2R$ complexes by measuring the stability of the backbone H-bond between Ile183[EL2.51] and Asn186[EL2.54], a key stabilizing force of the helix (*Figure 5a*). When we plotted the Ile183[EL2.51]-Asn186[EL2.54] distance against the Asp108[3.26]-Ile183[EL2.51] distance for each $D_2R$ complex (*Figure 5b*), we found that the loss of the Asp108[3.26]-Ile183[EL2.51] interaction increases the probability of breaking the Ile183[EL2.51]-Asn186[EL2.54] H-bond, that is the unwinding of EL2. Interestingly, in all our simulated $D_2R$ complexes, EL2 has a clear tendency to unwind, regardless of the scaffold of the bound ligand (*Figure 5c,d*, *Videos 1–3*). Note that in the $D_3R$/eticlopride simulations, the aligned residues Ser182[EL2.51] and Asn185[EL2.54] do not form such a H-bond, and EL2 is always in an extended conformation (*Figure 5b–d*). This tendency of EL2 to transition toward the extended conformation is also present in our simulations of $D_2R$ in complex with a partial agonist, aripiprazole, whereas EL2 in the $D_3R$ complexes with partial agonists (R22 and S22) remains in the extended conformation (*Table 1* and *Figure 5—figure supplement 3*). Interestingly, Asp104[3.26] and Ser182[EL2.51] can move into interacting range in the $D_3R$/eticlopride simulations, and the Ser182[EL2.51]-Asn185[EL2.54] interaction can sporadically form in the $D_3R$/R22 simulations – both raise the possibility that the extended conformation of $D_3R$ EL2 may transition to a helical conformation.

Interestingly, in one of our long MD trajectories of the $D_2R$/risperidone complex, EL2 evolved into a conformation that has a helical N-terminal portion and an extended C-terminal portion (*Video 4* and *Figure 5—figure supplement 4*). This conformation is not observed in either of the $D_2R$/risperidone and $D_3R$/eticlopride structures but is similar to that of the 5-HT$_{2A}$R/risperidone structure, further demonstrating the dynamics of this loop region (*Figure 5—figure supplement 4*).

In marked contrast to the obvious trend toward unwinding of EL2 in all our simulated $D_2R$ complexes, in our recent simulations of MhsT, a transporter protein with a region found by crystallography to alternate between helical and unwound conformations (*Malinauskaite et al., 2014*), we failed to observe any spontaneous unwinding over a similar simulation timescale

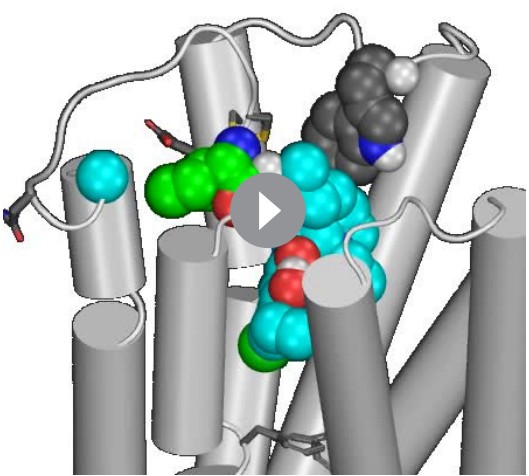

**Video 2.** A movie of a 4.2 µs $D_2R$/eticlopride trajectory shows the dynamics of Trp100[EL1.50] when the C-terminal portion of EL2 is in a helical conformation. Note that Trp100[EL1.50] can be stabilized by interacting with the disulfide bond. The presentation and color scheme are similar to those in *Video 1*, except that the bound carbon atoms of the ligand eticlopride are colored in cyan.
https://elifesciences.org/articles/52189#video2

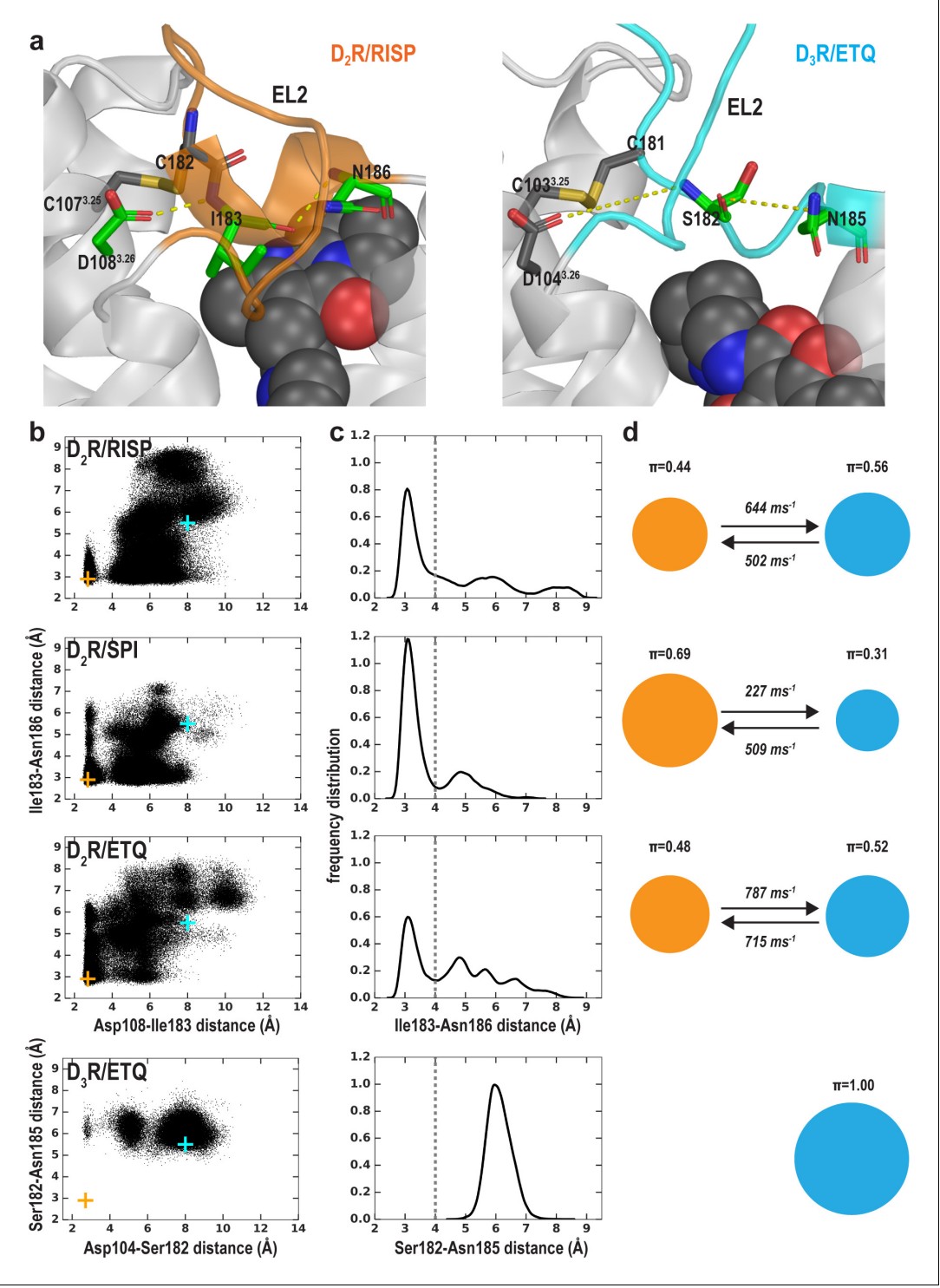

**Figure 5.** The helical conformation of EL2 in the $D_2R$/risperidone structure has a tendency to unwind in our simulations, regardless of the bound ligand. (a) The Ile183$^{EL2.51}$-Asn186$^{EL2.54}$ backbone H-bond and the Ile183$^{EL2.51}$-Asp108$^{3.26}$ interaction in $D_2R$ and their aligned interactions in $D_3R$. (b) The scatter plots of the two distances in the indicated $D_2R$ and $D_3R$ complexes. The orange and cyan crosses indicated the distances in the $D_2R$/risperidone and $D_3R$/eticlopride structures, respectively. (c) The distributions of the EL2.51-EL2.54 distances in the indicated simulations. These distances were used to evaluate the tendency to unwind using Markov state model (MSM) analysis in d). (d) The MSM analysis of the transition between the helical and extended conformational states of EL2. The area of each disk representing a state is proportional to the equilibrium

*Figure 5 continued on next page*

*Figure 5 continued*

probability (π) in each simulated condition. The values from the maximum likelihood Bayesian Markov model for π and transition rates from 500 Bayesian Markov model samples are shown. Thus, EL2 in all the $D_2R$ complexes show significant tendencies to unwind, while that in $D_3R$/eticlopride remains extended.

The online version of this article includes the following figure supplement(s) for figure 5:

**Figure supplement 1.** Sequence alignment and residue indices of EL1 and EL2 for the receptors being compared in this study.

**Figure supplement 2.** The helical region of EL2 of $D_2R$ can spontaneously unwind to an extended conformation similar to that of $D_3R$.

**Figure supplement 3.** The MSM analysis of Ile183-Asn186 distance in the simulations of the $D_2R$/aripiprazole, $D_3R$/S22, and $D_3R$/R22 complexes (*Table 1*).

**Figure supplement 4.** The distinct $D_2R$ EL2 conformations revealed by the MD simulations are similar to those of homologous receptors.

**Figure supplement 5.** The accessibility pattern of EL2 revealed by previous SCAM studies in $D_2R$ is more consistent with an extended EL2 conformation similar to that in the $D_3R$/eticlopride structure.

**Figure supplement 6.** Implied timescales (ITS) for the MSM analysis.

---

(with the longest simulations being ~5–6 μs) when the region was started from the helical conformation (*Abramyan et al., 2018*; *Stolzenberg et al., 2017*). This shows how difficult it can be to capture known dynamics in simulations and suggests that the C-terminal helical conformation of EL2 in $D_2R$ represents a higher energy state than the extended conformation, which allows for observation of the transitions in a simulation timescale not usually adequate to sample folding/unfolding events (*Piana et al., 2011*).

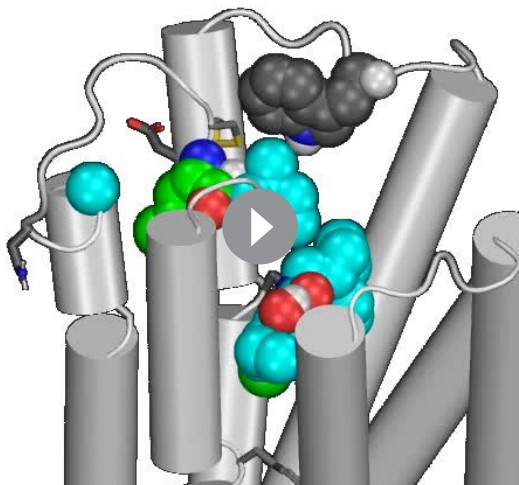

**Video 3.** A movie of a 3.6 μs $D_2R$/eticlopride trajectory collected using the CHARMM36 force field shows another example of unwinding of EL2. Thus, considering the similar unwinding pathway as that in *Video 1* (*Figure 5—figure supplement 2*), the unwinding does not depend on the force field used in the simulations or the identity of the antagonist bound in the OBS. Note the sidechain of Asn175$^{EL2.39}$ rotates inward and approaches Asp108$^{3.26}$ in this trajectory. The presentation and color scheme are the same as those in *Video 2*.

https://elifesciences.org/articles/52189#video3

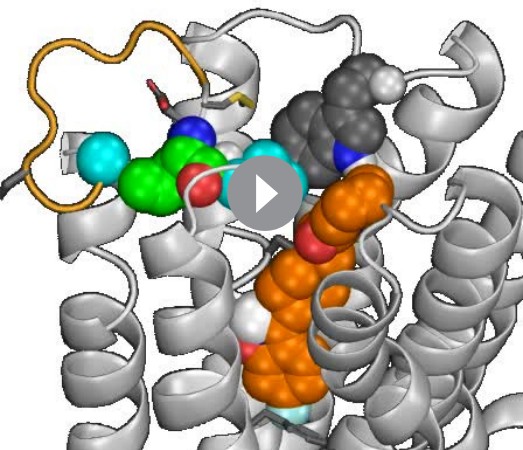

**Video 4.** A movie of a 4.5 μs $D_2R$/risperidone trajectory shows the N-terminal portion of EL2 can transition into a helical conformation when the C-terminal portion is extended. This is a novel EL2 conformation that has not been revealed by the $D_2R$, $D_3R$ or $D_4R$ structures but similar to those in the 5-HT$_{2A}$R/risperidone (*Figure 5— figure supplement 4f*), β$_1$AR and β$_2$AR structures. The presentation and color scheme are the same as those in *Video 1*.

https://elifesciences.org/articles/52189#video4

## Both the EL2 conformation and ligand scaffold affect the EL1 conformation

We have previously shown that the divergence in both the length and number of charged residues in EL1 among $D_2R$, $D_3R$, and $D_4R$ is responsible for the selectivity of more extended ligands (*Michino et al., 2013*; *Newman et al., 2012*). Another striking difference in the $D_2R$, $D_3R$, and $D_4R$ structures is the position of the conserved $Trp^{EL1.50}$ in EL1. $Trp100^{EL1.50}$ is in a much more inward position in the $D_2R$ structure, making a direct contact with the bound risperidone (*Figure 6a*), $Trp101^{EL1.50}$ in $D_4R$ interacts with the bound nemonapride that has an extended structure, whereas $Trp96^{EL1.50}$ in $D_3R$ is not in contact with eticlopride (*Figure 6b*). Thus, we asked whether these distinct positions of $Trp^{EL1.50}$ are due to the divergence in EL1 among these receptors (*Michino et al., 2013*) or due to the multiple inactive conformations that differentially accommodate the binding of non-selective ligands of divergent scaffolds.

When residues $182^{EL2.50}$-$186^{EL2.54}$ of EL2 are in a helical conformation, in the $D_2R$/risperidone simulations, we found that there is more room in the extracellular vestibule and the position of $Trp100^{EL1.50}$ is flexible and can adopt several positions and orientations (*Figure 6c,e,f*). In the $D_2R$/eticlopride simulations, $Trp100^{EL1.50}$, which cannot interact with eticlopride, shows more flexibility than that observed in the presence of risperidone and can move to a similar position like that of $Trp96^{EL1.50}$ in the $D_3R$ structure (*Figure 6—figure supplement 1* and *Video 2*). Interestingly, in this position, the conformation of $Trp^{EL1.50}$ can be stabilized by the disulfide bond of EL2 (*Ioerger et al., 1999*) (as shown in *Video 2*) or by interaction with the N terminus, which was truncated in the receptor construct used in the determination of the crystal structure. In the $D_2R$/spiperone simulations, the phenyl substitution on the triazaspiro[4.5]decane moiety protrudes toward the interface between TM2 and TM3, and contacts $Trp100^{EL1.50}$, which is flexible as well and can adopt a position that is even further away from the OBS than that of $Trp96^{EL1.50}$ in the $D_3R$ structure (*Figure 6—figure supplement 1*).

In contrast, when EL2 is in an extended conformation like that in $D_3R$, it restricts the flexibility of $Trp100^{EL1.50}$ (*Video 3*). This trend is consistent with the $D_3R$/eticlopride simulations in which we do not observe any significant rearrangement of $Trp96^{EL1.50}$ (*Figure 6d,e,f*).

Thus, we infer that the distinct conformation of $Trp100^{EL1.50}$ in the $D_2R$ structure is a combined effect of the helical EL2 conformation and the favored interaction that $Trp100^{EL1.50}$ can form with the bound risperidone in the crystal structure, the latter of which however, has a limited influence on the binding affinity of risperidone (*Wang et al., 2018*), consistent with the unstable interaction between risperidone and $Trp100^{EL1.50}$ in our simulations (*Figure 6*, *Video 2*). Indeed, in the fully extended EL2 conformation in which $Ile183^{EL2.51}$ rotates to face the extracellular vestibule, $Ile183^{EL2.51}$ makes a direct contact with the bound risperidone, whereas $Trp100^{EL1.50}$ loses its interaction with the ligand entirely (*Video 1*). Nevertheless, risperidone retains all other contacts in the OBS. In the recently reported $5\text{-}HT_{2A}R$/risperidone structure (PDB: 6A93) *Kimura et al. (2019)*, risperidone has a very similar pose in the OBS as that in the $D_2R$ structure, occupying the $Ile^{3.40}$ subpocket as well. However, on the extracellular side of the OBS, EL2 in the $5\text{-}HT_{2A}R$/risperidone complex is in an extended conformation and the EL2 residue $Leu228^{EL2.51}$ contacting risperidone aligns to $Ile183^{EL2.51}$ of $D_2R$, whereas the conserved $Trp141^{EL1.50}$ does not interact with risperidone in the $5\text{-}HT_{2A}R$. It is tempting to speculate that the EL2 and EL1 dynamics we observe in the $D_2R$/risperidone simulations represents a more comprehensive picture, as the divergent interactions shown in the extracellular loops of the $5\text{-}HT_{2A}R$/risperidone and $D_2R$/risperidone structures may not result from differences in the protein sequences of this dynamic region between these two receptors but rather two different static snapshots due to differences in the crystallographic conditions (Note risperidone has similarly high affinities for both $D_2R$ and $5HT_{2A}R$; *Kimura et al., 2019*; *Wang et al., 2018*).

Thus, the plasticity of the OBS and the dynamics of the extracellular loops appear to be two relatively separated modules in ligand recognition. To the extent of our simulations, we did not detect strong ligand-dependent bias in the EL2 dynamics as we did for the OBS. However, when EL2 is helical, the EL1 dynamics are sensitive to the bound ligand (compare *Figure 6* and *Figure 6—figure supplement 1*); when EL2 is extended, it restricts EL1 dynamics (*Figure 6*).

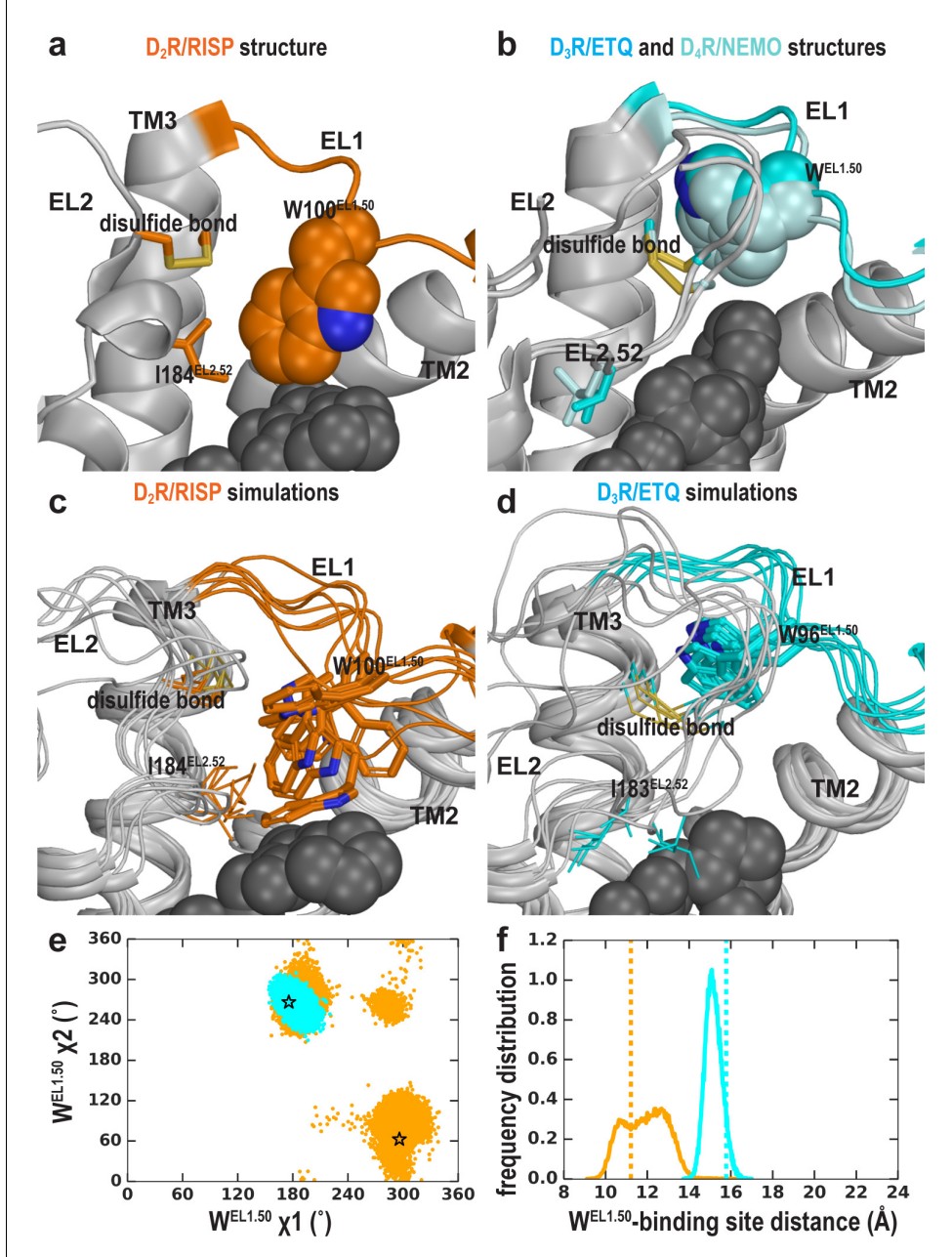

**Figure 6.** The EL2 conformation affects the EL1 conformation. Divergent EL1-EL2 interfaces among the $D_2R$ (**a**), $D_3R$, and $D_4R$ (**b**) structures. In the $D_2R$ structure, the Trp100$^{EL1.50}$ in EL1 forms a weak interaction with Ile184$^{EL2.52}$; while the aligned Trp96$^{EL1.50}$ of $D_3R$ and Trp101$^{EL1.50}$ in $D_4R$ are stabilized by their interactions with the disulfide bond – their passages toward the position of Trp100$^{EL1.50}$ in $D_2R$ are blocked by the extended EL2. In our simulations, Trp100$^{EL1.50}$ in $D_2R$ shows significant flexibility and can adopt multiple positions and orientations in $D_2R$/risperidone (**c**), while Trp96$^{EL1.50}$ in $D_3R$ is highly stable in $D_3R$/eticlopride (**d**). (**e**) The χ1 and χ2 dihedral angles of Trp100$^{EL1.50}$ in the subset of the $D_2R$/risperidone simulations in which EL2 is still in a helical conformation (orange), are more widely distributed than those of Trp96$^{EL1.50}$ in the $D_3R$/eticlopride simulations in which EL2 remains in extended conformations (cyan). These dihedral angle values in the $D_2R$ and $D_3R$ structures are indicated with the orange and cyan stars, respectively. (**f**), For the same two sets of simulations in **e**, the distance between the center of mass (COM) of the sidechain heavy atoms of Trp100 in $D_2R$ and the COM of the Cα atoms of the ligand-binding site residues (excluding Trp100, see Materials and methods for the list of the residues) has wider distributions than the corresponding distance between Trp96$^{EL1.50}$ in $D_3R$ and its ligand binding site. These distances in the $D_2R$ and $D_3R$ structures are indicated with the orange and cyan dotted lines, respectively.

*Figure 6 continued on next page*

*Figure 6 continued*

The online version of this article includes the following figure supplement(s) for figure 6:

**Figure supplement 1.** EL1 is dynamic in the $D_2R$/eticlopride and $D_2R$/spiperone simulations when EL2 is helical.

**Figure supplement 2.** Trp$^{EL1.50}$ is closely associated with Leu$^{2.64}$ regardless of the EL2 conformation.

## The Ile184$^{EL2.52}$-Trp100$^{EL1.50}$ interaction is not critical for risperidone binding

To further investigate the dynamics and coordination of EL2 and EL1 loops, we mutated Leu94$^{2.64}$, Trp100$^{EL1.50}$, and Ile184$^{EL2.52}$, and evaluated the effects of the L94A, W100A, and I184A, mutations on the binding affinities of eticlopride, risperidone, and spiperone. As shown in *Figure 6—figure supplement 2*, Leu94$^{2.64}$ and Trp100$^{EL1.50}$ are closely associated in both the $D_2R$ and $D_3R$ structures, while Ile184$^{EL2.52}$ interacts with Trp100$^{EL1.50}$ only in the $D_2R$ structure. In our time-resolved energy transfer (Tr-FRET) binding experiments, using a fluorescently labeled spiperone derivate (spiperone-d2) as a tracer ligand, we found that both L94A and W100A significantly reduced the affinities of all tested antagonists, whereas I184A only reduced the affinity of eticlopride while it improved that of risperidone (*Table 3*). Thus, the effects of the L94A and W100A mutations have similar trends, which appear independent of the effect of I184A. Indeed, for Trp100 to switch between the positions in the $D_2R$ and $D_3R$ structures, it must pass the steric hindrance of the sidechain of Leu94; thus, some effects of the L94A mutation may reflect its perturbation of the positioning of Trp100, and vice versa.

These findings support our conclusions that the close interaction between Ile184$^{EL2.52}$ and Trp100$^{EL1.50}$ revealed by the $D_2R$/risperidone crystal structure is not necessary for the stabilization of the risperidone pose. Indeed, in our simulations, EL2 has significant intrinsic dynamics and transitions from the helical to unwound conformation independent of the bound ligands (see above). When it is in an extended conformation, Ile184 is dissociated from Trp100.

## The clustering of the binding site conformations

Virtual screening has been widely used as an initial step in drug discovery for novel ligand scaffolds. To this end, we found that $D_2R$ can significantly change its binding site shape to accommodate antagonists bearing different scaffolds, while EL2 is intrinsically dynamic. Thus, it is necessary to comprehensively consider the binding site conformations in virtual screening campaigns against $D_2R$, because limiting the screening to only a single conformation will miss relevant ligands. Indeed,

**Table 3.** The effect of mutations on the binding affinities of selected $D_2R$ ligands as determined in Tr-FRET-binding experiments. The affinities of the fluorescently labeled spiperone derivative (Spiperone-d2) or unlabeled antagonists were determined in saturation experiments at WT or mutant SNAP-tagged $D_{2S}Rs$ stably expressed in FlpIn CHO cells. Binding affinity values for risperidone and eticlopride were obtained in competition binding experiments. Means of n independent experiments are shown with 95% confidence intervals (CIs).

| | Spiperone-d2 saturation binding | | | Spiperone-d2 competition binding | | | | | | | | |
|---|---|---|---|---|---|---|---|---|---|---|---|---|
| | | | | Eticlopride | | | Risperidone | | | Spiperone | | |
| SNAP-$D_{2S}R$ | $pK_d$ ($K_d$, nM) (95% CI) | N | Mut/ WT | $pK_i$ ($K_i$, nM) (95% CI) | N | Mut/ WT | $pK_i$ ($K_i$, nM) (95% CI) | N | Mut/ WT | $pK_i$ ($K_i$, nM) (95% CI) | N | Mut/ WT |
| WT | 8.54 (2.88) (8.32–8.77) | 9 | 1.0 | 10.06 (0.09) (9.90–10.21) | 8 | 1.0 | 8.47 (3.34) (8.15–8.80) | 7 | 1.0 | 9.96 (0.11) (9.76–10.18) | 8 | 1.0 |
| L94A | 7.71 (19.5) (7.41–8.00)* | 5 | 6.8 | 9.08 (0.83) (8.91–9.23)* | 4 | 9.2 | 8.02 (9.54) (7.86–8.17)* | 5 | 2.9 | 8.36 (4.37) (8.21–8.50)* | 5 | 39.7 |
| W100A | 7.39 (40.7) (7.21–7.56)* | 9 | 14.1 | 8.06 (8.71) (7.78–8.32)* | 4 | 96.8 | 7.60 (25.1) (7.41–7.79)* | 7 | 7.5 | 8.39 (4.07) (8.19–8.59)* | 7 | 37.0 |
| I184A | 8.79 (1.62) (8.58–9.00) | 5 | 0.6 | 9.34 (0.45) (8.94–9.75)* | 4 | 5 | 9.33 (0.47) (9.18–9.48)* | 5 | 0.1 | 9.78 (0.17) (9.51–10.05) | 5 | 1.6 |

*=significantly different from WT value, p<0.05, one-way ANOVA with Dunnett's post-hoc test.

the strategy of ensemble docking, in which each ligand is docked to a set of receptor conformers, has been adapted in recent virtual screening efforts (*Amaro et al., 2018*).

To characterize the OBS conformational ensemble sampled by $D_2R$ in complex with ligands bearing different scaffolds in the context of EL2 dynamics, we clustered the OBS conformations in our representative $D_2R$/eticlopride and $D_2R$/risperidone MD trajectories in which EL2 transitioned from helical to unwound conformations (see Materials and methods). As expected, the OBS conformations in these two complexes are significantly different and can be easily separated into distinct clusters. For the clustering results shown in *Table 4*, the average pairwise RMSDs of the OBS residues (apRMSDs, see Materials and methods) between the $D_2R$/eticlopride and $D_2R$/risperidone clusters are >1.1 Å, which are similar to that between the $D_2R$ and $D_3R$ structures (1.2 Å), while the apRMSDs within each cluster is smaller than those between any two clusters (*Figure 7*). Interestingly, at this level of clustering, when the two clusters for each complex are ~0.8–0.9 Å apRMSD away from each other, the extended and helical conformations of EL2 are always mixed in a cluster (*Table 4*). This observation suggests that the helical versus extended EL2 conformations are not closely associated with the OBS conformations.

Thus, while the centroid frames from each cluster can form an ensemble for future virtual screening for the primary scaffold occupying the OBS, in order to discover novel extended ligands that protrude out of the OBS to interact with EL2 and EL1 residues (*Michino et al., 2015a*), additional frames that cover both helical and extended EL2 conformations from each cluster will have to be used to screen for the optimal extensions of the primary scaffold.

## Discussion

Our results highlight unappreciated conformational complexity of the inactive state of GPCRs and suggest that the risperidone bound $D_2R$ structure represents only one of a number of possible inactive conformations of $D_2R$. Critically, this conformation is incompatible with the binding of other high-affinity $D_2R$ ligands such as eticlopride. While distinct conformational states responsible for functional selectivity have garnered great attention, the potential existence of divergent inactive conformations is of critical importance as well. By combining in silico and in vitro findings, we propose that occupation of the $Ile^{3.40}$ sub-pocket by antagonists confers a distinct $D_2R$ conformation that is associated with both a greater degree of inverse agonism and $Na^+$ insensitivity in binding, such that $Na^+$ sensitivity is negatively related with the extent of inverse agonism for the tested ligands. However, other structural elements may also contribute to the extent of inverse agonism (*Zhang et al., 2014*). Regardless, the distinct inactive conformations stabilized by antagonists with different scaffolds may reflect different degrees of inactivation.

In addition to advancing our mechanistic understanding of receptor function, our findings have implications for high-throughput virtual screening campaigns, as important hits would be missed by focusing on a single inactive state captured in a crystal structure that is stabilized by an antagonist bearing a specific scaffold. Moreover, rational lead optimization requires rigorous physical

**Table 4.** Clustering results of the OBS conformations sampled in the $D_2R$/eticlopride and $D_2R$/risperidone simulations.

The compositions in each cluster are shown as percentages of the frames randomly extracted for each complex (see Materials and methods), when sorted by either receptor/ligand complex or EL2 conformation.

| | Percentage (%) | | | | | | | |
| --- | --- | --- | --- | --- | --- | --- | --- | --- |
| | Complex | | | | EL2 conformation | | | |
| | $D_2R$/eticlopride | | $D_2R$/risperidone | | Extended | | Helical | |
| Cluster ID | Mean | Sd | Mean | Sd | Mean | Sd | Mean | Sd |
| 1 | 38.4 | 0.7 | 0.0 | 0.0 | 4.9 | 0.4 | 33.5 | 0.5 |
| 2 | 61.6 | 0.7 | 0.0 | 0.0 | 45.1 | 0.4 | 16.5 | 0.6 |
| 3 | 0.0 | 0.0 | 43.7 | 1.0 | 2.5 | 0.4 | 41.3 | 0.8 |
| 4 | 0.0 | 0.0 | 56.3 | 1.0 | 47.5 | 0.4 | 8.7 | 0.8 |

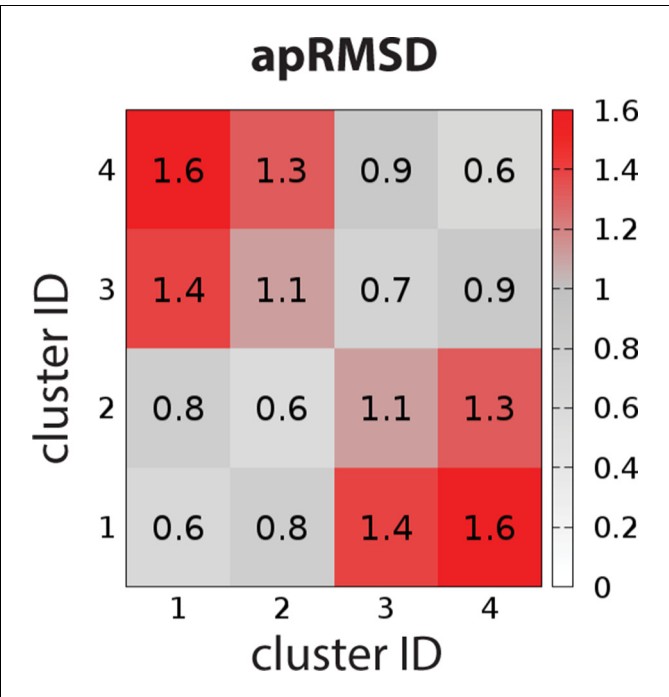

**Figure 7.** The average pairwise RMSDs of the clusters of the OBS conformations. The clustering level was chosen to be 4, so that the average pairwise RMSDs (apRMSDs) between the $D_2R$/eticlopride clusters (1 and 2, see *Table 4* for the composition of each cluster) and $D_2R$/risperidone clusters (3 and 4) are similar to that between $D_2R$ and $D_3R$ structures (1.2 Å), while all the apRMSDs within a cluster are smaller than those between any given two clusters.

description of molecular recognition (*Beuming and Shi, 2017*), which depends on adequate understanding of the conformational boundary and flexibility of the targeted state. We have shown previously that both dopamine receptor subtype selectivity and modulation of agonist efficacy can be achieved through the design of ligands that extend from the OBS into an extracellular secondary binding pocket (SBP) (*Michino et al., 2015a*; *Newman et al., 2012*). We now show that one might consider the occupation of the Ile$^{3.40}$ sub-pocket in the process of decorating an $D_2R$ antagonist scaffold to attain a desired level of inverse agonism. Our findings also reveal allosteric communication between the Ile$^{3.40}$ sub-pocket and the Na$^+$-binding site. Thus, Na$^+$ sensitivity in antagonist binding may provide useful mechanistic insights as part of such efforts.

The mutation of Trp100$^{EL1.50}$ in $D_2R$ to alanine, leucine or phenylalanine cause substantial increases in both the association and dissociation rate of risperidone (*Wang et al., 2018*). Curiously, both the dissociation and association rates of $D_2R$ antagonists used as antipsychotics have been proposed to determine their propensity to cause extrapyramidal side-effects and hyperprolactinaemia (*Seeman, 2014*; *Sykes et al., 2017*). Our results indicate that both the EL2 conformation and antagonist scaffolds may influence the dynamics of Trp100$^{EL1.50}$, which in turn controls ligand access and egress to and from the OBS. Thus, understanding the relationship between the distinct inactive $D_2R$ conformations stabilized by different antagonist scaffolds and these kinetic parameters will likely be important to facilitate the design of $D_2R$ antagonists with an optimal kinetic profile that minimizes the risk of side effects.

Previously, using the substituted-cysteine accessibility method (SCAM) in $D_2R$ (*Javitch et al., 2000*; *Shi and Javitch, 2004*), we found that G173$^{EL2.37}$C, N175$^{EL2.39}$C, and I184$^{EL2.52}$C were accessible to charged MTS reagents and that this accessibility could be blocked by the bound Na$^+$-sensitive antagonist sulpiride, consistent with their water accessibility and involvement in ligand binding and not with a static orientation facing lipid, whereas A177$^{EL2.45}$C and I183$^{EL2.51}$C were accessible but not protected by sulpiride. Curiously, in the $D_2R$/risperidone structure, Ile184$^{EL2.52}$ is only marginally in contact with the ligand, Ile183$^{EL2.51}$ blocks the accessibility of Gly173$^{EL2.37}$ to the OBS and

is itself buried in a hydrophobic pocket, whereas Asn175[EL2.39] faces lipid, where it would be much less reactive. In the $D_3R$/eticlopride structure, Ile183[EL2.52] is in close contact with the bound ligand, Ser182[EL2.51] faces the extracellular vestibule, whereas the sidechain of Asn173[EL2.39] is oriented toward the OBS (*Figure 5—figure supplement 5*). Thus, our analysis shows that the accessibility pattern of EL2 revealed by previous SCAM studies in $D_2R$ are more consistent with the extended EL2 conformation revealed by the $D_3R$/eticlopride structure but not with the $D_2R$/risperidone structure. Indeed, we observed spontaneous transitions of EL2 from a helical to extended conformation in our $D_2R$ simulations, which suggests that EL2 of $D_2R$ exists in an ensemble of structured and unwound conformations, with substantial occupation of the configuration found in the $D_3R$ structure. Such dynamics of EL2 suggest that the drastically different conformations between the $D_2R$ and $D_3R$ structures near EL2 are not related to the divergence of the receptors. Thus, the $D_2R$ EL2 appears to have quite dramatic dynamics that are not captured by the crystal structure.

Taken together, our findings reveal that both the plasticity of the transmembrane domain in accommodating different scaffolds and the dynamics of EL2 and EL1 are important considerations in RDD targeting the inactive conformation of $D_2R$.

## Materials and methods

**Key resources table**

| Reagent type (species) or resource | Designation | Source or reference | Identifiers | Additional information |
|---|---|---|---|---|
| Cell line (*Cricetulus griseus*) | FlpIn CHO | Invitrogen | Cat# R75807 | |
| Transfected construct (human) | SNAP-$D_{2S}R$ | Cisbio | Cat# pSNAPD2 | |
| Transfected construct (human) | $D_2R$ $G\alpha_{oA}$-RLuc8 $G\beta1$ $G\gamma2$-Venus | *Michino et al., 2017* | N/A | |
| Commercial assay or kit | Spiperone-d2 SNAP-Lumi4-Tb 5x SNAP/CLIP labeling medium | Cisbio | Cat# L0002RED Cat# SSNPTBX Cat# LABMED | |
| Chemical compound, drug | Na bisulfite Glucose (+)-Butaclamol Risperidone Haloperidol | Sigma Aldrich | Cat# 243973 Cat# D9434 Cat# D033 Cat# R3030 Cat# H1512 | |
| Chemical compound, drug | Spiperone | Cayman chemicals | Cat# 19769 | |
| Chemical compound, drug | Eticlopride HCl Raclopride (-)-Sulpiride Quinpirole | Tocris Bioscience | Cat# 1847 Cat# 1810 Cat# 0895 Cat# 1061 | |
| Chemical compound, drug | [³H]spiperone | Perkin Elmer | Cat# NET1187250UC | |
| Chemical compound, drug | Polyethylenimine | Polysciences | Cat# 23966 | |
| Chemical compound, drug | Coelenterazine-h | NanoLight Technology | Cat# 301–5 | |
| Software, algorithm | Prism | GraphPad | v7.0 and v8.2.1 | |

### Residue indices in EL1 and EL2

Based on a systematic analysis of aminergic receptors, we found a Trp in the middle of EL1 and the disulfide-bonded Cys in the middle of EL2 are the most conserved residues in each segment, and defined their residue indices as EL1.50 and EL2.50, respectively (*Michino et al., 2015a*), In this study, for the convenience of comparisons among $D_2R$, $D_3R$, and $D_4R$, and 5-$HT_{2A}R$, based on the

alignments of EL1 And EL2 shown in *Figure 5—figure supplement 1*, we index the EL1 and EL2 residues of each receptor in the same way as the Ballesteros-Weinstein numbering, for example the residues before and after the EL2.50 are EL2.49 and EL2.51, respectively. Note the indices for the shorter sequences are not necessarily be consecutive, given the gaps in the alignment.

## Molecular modeling and docking

The $D_2R$ models in this study are based on the corrected crystal structure of $D_2R$ bound to risperidone (PDB: 6CM4) (*Wang et al., 2018*). We omitted T4 Lysozyme fused into intracellular loop 3. Three thermostabilizing mutations (Ile122$^{3.40}$A, L375$^{6.37}$A, and L379$^{6.41}$A) were reverted to their WT residues. The missing N terminus in the crystal structure was built de novo using Rosetta (*Bradley et al., 2005*), and then integrated with the rest of the $D_2R$ model using Modeller (*John and Sali, 2003*). Using Modeller, we also extended two helical turns at the TM5 C terminus and three residues at the TM6 N terminus of the structure and connected these two ends with a 9 Gly loop, similar to our experimentally validated treatment of D3R models (*Michino et al., 2017*). The position of the $Na^+$ bound in the canonical $Na^+$-binding site near the negatively charged Asp$^{2.50}$ was acquired by superimposing the $Na^+$-bound structure of adenosine $A_{2A}$ receptor (*Liu et al., 2012*) to our $D_2R$ models.

The binding poses of risperidone and eticlopride were taken according to their poses in the $D_2R$ (*Wang et al., 2018*) and $D_3R$ (*Chien et al., 2010*) structures, respectively. Docking of spiperone in our D2R model was performed using the induced-fit docking (IFD) protocol (*Sherman et al., 2006*) in the Schrodinger software (release 2017–2; Schrodinger, LLC: New York NY). Based on our hypothesis regarding the role of the Ile$^{3.40}$ sub-pocket in the $Na^+$ sensitivity (see text), from the resulting poses of IFD, we choose the spiperone pose with the F-substitution on the butyrophenone ring occupying the Ile$^{3.40}$ sub-pocket. Note that in risperidone and spiperone the F-substitutions have similar distances to the protonated N atoms that interact with Asp$^{3.32}$ (measured by the number of carbon atoms between them, *Figure 1—figure supplement 1*).

## Molecular dynamics (MD) simulations

MD simulations of the $D_2R$ and $D_3R$ complexes were performed in the explicit water and 1-palmitoyl-2-oleoylphosphatidylcholine (POPC) lipid bilayer environment using Desmond MD System (version 4.5; D. E. Shaw Research, New York, NY) with either the OPLS3e force field (*Roos et al., 2019*) or the CHARMM36 force field (*Best et al., 2012*; *Klauda et al., 2010*; *MacKerell et al., 1998*; *MacKerell et al., 2004*) and TIP3P water model. For CHARMM36 runs, the eticlopride parameters were obtained through the GAAMP server (*Huang and Roux, 2013*), with the initial force field based on CGenFF assigned by ParamChem (*Vanommeslaeghe et al., 2010*). The system charges were neutralized, and 150 mM NaCl was added. Each system was first minimized and then equilibrated with restraints on the ligand heavy atoms and protein backbone atoms, followed by production runs in an isothermal–isobaric (NPT) ensemble at 310 K and one atm with all atoms unrestrained, as described previously (*Michino et al., 2017*; *Michino et al., 2015b*). We used Langevin constant pressure and temperature dynamical system (*Feller et al., 1995*) to maintain the pressure and the temperature, on an anisotropic flexible periodic cell with a constant-ratio constraint applied on the lipid bilayer in the X-Y plane. For each condition, we collected multiple trajectories, the aggregated simulation length is ~392 μs (*Table 1*).

While the majority of our $D_2R$ simulations in this study used the OPLS3e force field, to compare with the $D_3R$ simulations using CHARMM36 that have been continued from the previously reported shorter trajectories (*Michino et al., 2017*; *Michino et al., 2015b*), we carried out the $D_2R$/eticlopride simulations using both the OPLS3e and CHARMM36 force fields (see *Table 1*). We did not observe significant differences and pooled their results together for the analysis.

## Conformational analysis

Distances and dihedral angles of MD simulation results were calculated with MDTraj (version 1.8.2) (*McGibbon et al., 2015*) in combination with *in-house* Python scripts.

To characterize the structural changes in the receptor upon ligand binding, we quantified differences of structural elements between the $D_2R$/eticlopride and $D_2R$/risperidone conditions (using last 600 ns from a representative trajectory for each condition), by applying the previously described

pairwise interaction analyzer for GPCR (PIA-GPCR) (*Michino et al., 2017*). The subsegments on the extracellular side of $D_2R$ were defined as following: TM1e (the extracellular subsegment (e) of TM1, residues 31–38), TM2e (residues 92–96), TM3e (residues 104–113), TM4e (residues 166–172), TM5e (residues 187–195), TM6e (residues 364–369), and TM7e (residues 376–382).

For the PIA-GPCR analysis in *Figure 4* and the distance analysis in *Figure 6*, we used the set of ligand-binding residues previously identified by our systematic analysis of GPCR structures. Specifically, for $D_2R$, they are residues 91, 94, 95, 100, 110, 111, 114, 115, 118, 119, 122, 167, 184, 189, 190, 193, 194, 197, 198, 353, 357, 360, 361, 364, 365, 367, 368, 376, 379, 380, 383, 384, 386, and 387; for $D_3R$, they are residues 86, 89, 90, 96, 106, 107, 110, 111, 114, 115, 118, 165, 183, 188, 189, 192, 193, 196, 197, 338, 342, 345, 346, 349, 350, 352, 353, 362, 365, 366, 369, 370, 372, and 373.

For the clustering of the OBS conformations, we used representative $D_2R$/eticlopride and $D_2R$/risperidone MD trajectories in which EL2 transitioned from the helical to unwound conformations. For each complex, using the Ile183-Asn186 distance as a criterion to differentiate the EL2 conformation (*Figure 5*), 1000 MD frames with helical EL2 conformations and another 1000 frames with extended EL2 conformations were randomly selected. For these 4000 frames, the pairwise RMSD of the backbone heavy atoms of the OBS residues defined in *Michino et al. (2015a)*, except for Ile184$^{EL2.52}$, were calculated. The resulting 4000 × 4000 matrix was used to cluster these frames using the k-mean algorithm implemented in R. We chose nstart to be 20 to assure the convergence of cluster centroids and boundaries. We chose the clustering level to be 4, so that the average pairwise RMSDs (apRMSDs) between the $D_2R$/eticlopride and $D_2R$/risperidone clusters are similar to that between $D_2R$ and $D_3R$ structures (1.2 Å), while all the apRMSDs within a cluster are smaller than those between any given two clusters. The same frame selection and clustering procedure was repeated to 20 times. The averages of these 20 runs for the compositions of each cluster were reported in *Table 4*.

## Markov State Model (MSM) analysis

The MSM analysis was performed using the pyEMMA program (version 2.5.5) (*Scherer et al., 2015*). To characterize the dynamics of EL2 of $D_2R$, specifically the transitions between helical and extended conformations of its C-terminal portion, we focused on a key hydrogen bond formed in the helical conformation between the backbone carbonyl group of Ile183 and the backbone amine group of Asn186. Thus, for each of the simulated conditions, the distance of Ile183-Asn186 (Ser182-Asn185 in $D_3R$) was used as an input feature for the MSM analysis. We discretized this feature into two clusters – distances below and above 4 Å (i.e. EL2 forming a helical conformation and unwinding). Implied relaxation timescale (ITS) (*Swope et al., 2004*) for the transition between these clusters was obtained as a function of various lag times. Convergences of ITS for the MSMs for all conditions was achieved at a lag time of 300 ns (*Figure 5—figure supplement 6*), which we further used to estimate Bayesian Markov models with 500 transition matrix samples (*Trendelkamp-Schroer and Noé, 2013*). The maximum likelihood transition matrix was used to calculate the transition and equilibrium probabilities ($\pi$) shown in *Figure 5* and *Figure 5—figure supplement 3*.

## Cell culture and cell line generation

Site-directed mutagenesis was performed using the Quickchange method using pEF5/DEST/FRT plasmid encoding FLAG-SNAP-$D_{2S}R$ as the DNA template. The mutagenesis was confirmed, and the full coding region was checked using Sanger sequencing at the DNA Sequencing Laboratory (University of Nottingham). Stable cell lines were generated using the Flp-In recombination system (Invitrogen).

## [³H]spiperone binding assay

FlpIn CHO cells (Invitrogen) stably expressing WT or mutant SNAP-D2s cells were cultured before the preparation of cell membrane as described before (*Klein Herenbrink et al., 2019*). All stable cell lines were confirmed to be mycoplasma free. For saturating binding assays cell membranes (Mutant or WT SNAP-$D_{2s}$-FlpIn CHO, 2.5 µg) were incubated with varying concentrations of [³H]spiperone and 10 µM haloperidol as a non-specific control, in binding buffer (20 mM HEPES, 100 mM NaCl, 6 mM MgCl₂, 1 mM EGTA, and 1 mM EDTA, pH 7.4) to a final volume of 200 µL and were incubated at 37°C for 3 hr. For competition binding assays, cell membranes (SNAP-$D_{2s}$-FlpIn CHO,

2.5 µg) were incubated with varying concentrations of test compound in binding buffer containing 0.2 nM of [$^3$H]spiperone to a final volume of 200 µL and were incubated at 37°C for 3 hr. Binding was terminated by fast-flow filtration using a Uniplate 96-well harvester (PerkinElmer) followed by five washes with ice-cold 0.9% NaCl. Bound radioactivity was measured in a MicroBeta2 LumiJET MicroBeta counter (PerkinElmer). Data were collected from at least three separate experiments performed in triplicate and analysed using non-linear regression (Prism 7, Graphpad software). For radioligand saturation binding data, the following equation was globally fitted to nonspecific and total binding data:

$$Y = \frac{B_{\max}[A]}{[A] + K_A} + NS[A] \tag{1}$$

where Y is radioligand binding, $B_{\max}$ is the total receptor density, [A] is the free radioligand concentration, $K_A$ is the equilibrium dissociation constant of the radioligand, and NS is the fraction of nonspecific radioligand binding. The $B_{\max}$ of the SNAP-tagged D2SRs we as follows; WT = 7.95 ± 1.63 pmol.mg$^{-1}$, 6.39 ± 1.04 pmol.mg$^{-1}$, 4.37 ± 0.92 pmol.mg$^{-1}$, 2.61 ± 0.50 pmol.mg$^{-1}$.

For competition binding assays, the concentration of ligand that inhibited half of the [$^3$H]spiperone binding (IC$_{50}$) was determined by fitting the data to the following equation:

$$Y = \frac{Bottom + (Top - Bottom)}{1 + 10^{(X - LogIC_{50})n_H}} \tag{2}$$

where Y denotes the percentage specific binding, Top and Bottom denote the maximal and minimal asymptotes, respectively, IC$_{50}$ denotes the X-value when the response is midway between Bottom and Top, and $n$H denotes the Hill slope factor. IC$_{50}$ values obtained from the inhibition curves were converted to $K_i$ values using the Cheng and Prusoff equation. No statistical methods were used to predetermine sample size.

## Bioluminescence resonance energy transfer (BRET) assay

The Go-protein activation assay uses a set of BRET-based constructs previously described (*Michino et al., 2017*). Briefly, HEK293T cells were transiently co-transfected with pcDNA3.1 vectors encoding (i) D$_2$R, (ii) G$\alpha_{oA}$ fused to Renilla luciferase 8 (Rluc8; provided by Dr. S. Gambhir, Stanford University, Stanford, CA) at residue 91, (iii) untagged Gβ1, and (iv) Gγ2 fused to mVenus. Transfections were performed using polyethyleneimine (PEI) at a ratio of 2:1 (PEI:total DNA; weight:weight), and cell culture was maintained as described previously (*Bonifazi et al., 2019*). After ~48 hr of transfection, cells were washed with PBS and resuspended in PBS + 0.1% glucose + 200 µM Na Bisulfite buffer. Approximately 200,000 cells were then distributed in each well of the 96-well plates (White Lumitrac 200, Greiner bio-one). 5 µM Coelenterazine H, a luciferase substrate for BRET, was then added followed by addition of vehicle and test compounds using an automated stamp transfer protocol (Nimbus, Hamilton Robotics) from an aliquoted 96-well compound plate. Following ligands were used – quinpirole, eticlopride, raclopride, and (-)-sulpiride (Tocris Bioscience), (+)-butaclamol, dopamine, and risperidone (Sigma Aldrich), and Spiperone (Cayman chemicals). mVenus emission (530 nm) over RLuc 8 emission (485 nm) were then measured after 30 min of ligand incubation at 37°C using a PHERAstar *FSX* plate reader (BMG Labtech). BRET ratio was then determined by calculating the ratio of mVenus emission over RLuc eight emission.

Data were collected from at least nine independent experiments and analyzed using Prism 7 (GraphPad Software). Drug-induced BRET, defined by BRET ratio difference in the presence and absence of compounds, was calculated. Concentration response curves (CRCs) were generated using a non-linear sigmoidal dose-response analyses using Prism 7 (GraphPad Software). CRCs are presented as mean drug-induced BRET ± SEM. E$_{max}$ bar graphs are plotted as the percentage of maximal drug-induced BRET by quinpirole ± SEM.

## Tr-FRET ligand binding

*Materials:* Spiperone-d2, SNAP-Lumi4-Tb and 5x SNAP/CLIP labeling medium were purchased from Cisbio Bioassays. Eticlopride hydrochloride was purchased from Tocris Bioscience. Saponin was purchased from Fluka/Sigma-Aldrich. Bromocriptine, haloperidol, risperidone, spiperone, pluronic-F127,

Gpp(NH)p, DNA primers, Hanks Balanced Salt Solution H8264 (HBSS) and phosphate buffered saline (PBS) was purchased from Sigma-Aldrich.

## Terbium cryptate labeling and membrane preparation

Terbium cryptate labeling of the SNAP-tagged receptors and membrane preparation was performed with minor changes to previously described methods (*Klein Herenbrink et al., 2016*). Flp-In CHO-K1 cells stably expressing the mutant SNAP-$D_{2S}$R constructs were grown in T175 flasks to approximately 90% confluency. Cell media was aspirated, and the cells were washed twice with 12 mL PBS. The cells were then incubated with terbium cryptate labeling reagent in 1xSNAP/CLIP labeling medium for 1 hr at in a humidified cell culture incubator with 5% $CO_2$ at 37°C. The terbium cryptate labeling reagent was then removed and the cells were washed once with 12 mL PBS. The labeled cells were then harvested in 10 mL PBS by cell scraping. Harvested cells were then collected by centrifugation at 300 g for 5 min and removal of the supernatant. The cell pellets were then frozen at −80°C for later membrane preparation. For cell membrane preparation, each cell pellet was removed from the −80°C freezer and thawed on ice. The pellet was then resuspended in 10 mL of ice-cold Buffer 1 (10mM HEPES 10 mM EDTA pH7.4). The pellet was then homogenised (IKA works T 10 basic Ultra-Turrax homogeniser) with eight bursts of 3 s on setting 4. The homogenized cells were transferred to an ultra-fast centrifge tube and an additional 10 mL of Buffer one was added. The tube was then centrifuged at 48,000 g for 30 min at 4°C. The supernatant was discarded, 20 mL of Buffer one was added and the pellet was resuspended. The resuspension was then centrifuged a second time at 48,000 g for 30 min at 4°C. The supernatant was then removed, and the cell membrane pellet was collected by resuspension in 2 mL ice-cold Buffer 2 (10mM HEPES 0.1 mM EDTA pH 7.4). The resuspended membranes were then put through a syringe with a BD precision glide 26-gauge needle to make the solution uniform. Membrane protein concentration was determined by bicinchonic acid (BCA) assay detecting the absorbance at 562 nm on a CLARIOstar plate reader (BMG Labtech) using bovine serum albumin (BSA) as the protein standard. The cell membrane solution was then aliquoted and frozen at −80°C.

## TR-FRET binding assay

All ligands were diluted in Binding Buffer (Hanks Balanced Salt Solution (Sigma H8264), 20 mM HEPES, 0.02% Pluronic-F127, 1% dimethyl sulfoxide, pH 7.4 (with KOH)). For competition binding experiments; 10 μL of spiperone-d2 in Binding Buffer was added to each well of a 384-well white optiplate LBS coated (PerkinElmer) at varied concentrations depending on the SNAP-$D_{2S}$R mutant. 10 μL of increasing concentrations of unlabeled ligands were then added into the 10 μL of fluorescent ligand and mixed. A final concentration of 100 μM haloperidol was used to determine non-specific binding. Cell membranes were diluted to 0.075 μg/μL in Binding Buffer supplemented with 50 μg/mL saponin and 100 μM Gpp(NH)p.

   TR-FRET measurements were acquired on a PHERAstar *FS* plate reader (BMG Labtech) at 37°C. The optiplate containing the ligand cocktails in the wells was incubated in the instrument for 6 min. The cell membrane solution was primed into the on-board injection system and incubated for 5 min. 20 μL of cell membrane solution was injected at 400 μL/s into the ligand cocktail wells to initiate the binding reaction. After 30-min incubation, the HTRF optic filter module was used to perform an excitation at 337 nm and simultaneous dual emission detection at 620 nm (terbium cryptate donor) and 665 nm (fluorescent ligand acceptor). The focal height was set to 10.4 mm. All experiments were performed in singlet wells. The TR-FRET binding values were determined by dividing the by the fluorescent ligand acceptor channel values by the terbium cryptate donor channel values and multiplying by 10,000. These values were then subtracted by the non-specific binding values determined in each experiment to give the specific HTRF ratio x 10,000. The data was then analysed with GraphPad Prism 8.2.1 using *Equations 1 and 2*.

## Acknowledgements

Support for this research was provided by the National Institute on Drug Abuse–Intramural Research Program, Z1A DA000606-03 (LS), NIH grant MH54137 (JAJ) and National Health and Medical

Research Council (NHMRC) Project Grant APP1049564 (JRL). We thank Jackie Glenn for technical support in generating membrane preparations.

## Additional information

### Funding

| Funder | Grant reference number | Author |
|---|---|---|
| National Institutes of Health | Z1A DA000606-03 | Lei Shi |
| National Institutes of Health | MH54137 | Jonathan A Javitch |
| National Health and Medical Research Council | APP1049564 | J Robert Lane |

The funders had no role in study design, data collection and interpretation, or the decision to submit the work for publication.

### Author contributions

J Robert Lane, Conceptualization, Data curation, Formal analysis, Supervision, Funding acquisition, Investigation; Ara M Abramyan, Pramisha Adhikari, Ravi Kumar Verma, Herman D Lim, Hideaki Yano, Data curation, Formal analysis, Investigation; Alastair C Keen, Kuo-Hao Lee, Julie Sanchez, Formal analysis, Investigation; Jonathan A Javitch, Conceptualization, Investigation; Lei Shi, Conceptualization, Data curation, Formal analysis, Supervision, Funding acquisition, Investigation, Project administration

### Author ORCIDs

Lei Shi ![ORCID] https://orcid.org/0000-0002-4137-096X

### Decision letter and Author response

Decision letter https://doi.org/10.7554/eLife.52189.sa1
Author response https://doi.org/10.7554/eLife.52189.sa2

## Additional files

### Supplementary files

• Transparent reporting form

### Data availability

All data generated or analysed during this study are included in the manuscript and supporting files.

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
