## [Decision Letter]

**Acceptance summary:**

Historically, a G Protein Coupled Receptor (GPCR) is believed to assume an active conformation or an inactive conformation that are both well defined. Using molecular dynamics simulations combined with binding and functional assays, in this study the author showed that the inactive conformation of eticlopride-bound dopamine receptor D2 differs from that of risperidone-bound D2, and the conformational heterogeneity many underlie the different peak potency of these two inverse agonists. For other protein systems such as protein kinases it is well known that the active conformations are highly similar while the inactive conformations can vary greatly, but for GPCRs this is still a relatively new concept, which opens new possibilities for structural-based drug designs targeting GPCRs.

**Decision letter after peer review:**

Thank you for submitting your article "Distinct inactive conformations of the dopamine D2 and D3 receptors correspond to different extents of inverse agonism" for consideration by *eLife*. Your article has been reviewed by three peer reviewers, one of whom is a member of our Board of Reviewing Editors, and the evaluation has been overseen by Richard Aldrich as the Senior Editor. The reviewers have opted to remain anonymous.

The reviewers have discussed the reviews with one another and the Reviewing Editor has drafted this decision to help you prepare a revised submission.

Summary:

The authors performed extensive MD simulation starting with existing crystal structure of D2 receptor with different non-subtype selective antagonist. The authors identified different conformation observed in D2, D3, D4 receptors that were not due to differences in the receptors themselves, but result from different antagonists used in crystallization. Sodium ion modulation on dopamine antagonist activity has been reported previously, however, the underlying mechanism was unclear. In this study, the authors proposed that the sodium insensitive antagonist occupies a sub-pocket near the sodium binding site, whereas the sodium sensitive antagonist does not. Furthermore, the occupation of the sub-pocket is suggested to be correlated with the strength of inverse agonism of the ligand. The authors also found EL2 of D2 receptor transition from helical conformation to extended conformation and these conformations are not ligand dependent. The additional information of the receptor conformation from MD studies may well be useful for ligand design.

Essential revisions:

1) A key supporting set of biochemical data is depicted in Figure 3, where Na^+^-sensitive ligands are shown to be weaker inverse agonists than Na^+^-insensitive ligands. This finding seems inconsistent with some prior work on D_2_R inverse agonists. (See several citations in Zhang et al., 2014) It undermines the significance of this work if this observation is true only for the specific assay conditions in the present manuscript. At a minimum, this apparent discrepancy needs to be discussed in the manuscript. Author has utilized BRET assay that measure distance between Gα and Gγ subunits to demonstrate different level of inverse agonism at D2 receptor. But one reviewer points out that the distance between two probes measured by BRET may not be a good indicator of inverse agonism. Functional assay like inhibition of cAMP or [^35^S] GTPγS assay would be more appropriate for this purpose. Was the BRET assay performed in the presence of sodium ion?

2) The MD findings related to EL1 and EL2 lacks adequate support from biochemical experiments. Testing a few mutations related to the EL1 and EL1 findings would strengthen this manuscript.

3) Four ligands that occupy the Ile^3.40^ sub-pocket are docked, but only one that does not occupy the pocket. Given that Nemonapride is also a non-selective and high affinity D_2_R antagonist, this ligand should be included in the MD study.

4) With regards to the different conformation identified in EL2, could the author comment on which conformation identified will likely be useful for virtual screening in drug discovery?

5) Subsection “Both the EL2 conformation and ligand scaffold affect the EL1 conformation”, second paragraph. Interestingly, in this position, the conformation of Trp^EL1.50^ can be stabilized by the disulfide bond of EL2 (Ioerger et al., 1999) (as shown in Video 2) or by interaction with the N terminus, which was truncated in the receptor construct used in the determination of the crystal structure. – Is there any evidence for the interaction with the N-terminus?

---

## [Author Response]

Essential revisions:1) A key supporting set of biochemical data is depicted in Figure 3, where Na^+^-sensitive ligands are shown to be weaker inverse agonists than Na^+^-insensitive ligands. This finding seems inconsistent with some prior work on D_2_R inverse agonists. (See several citations in Zhang et al., 2014) It undermines the significance of this work if this observation is true only for the specific assay conditions in the present manuscript. At a minimum, this apparent discrepancy needs to be discussed in the manuscript. Author has utilized BRET assay that measure distance between Gα and Gγ subunits to demonstrate different level of inverse agonism at D2 receptor. But one reviewer points out that the distance between two probes measured by BRET may not be a good indicator of inverse agonism. Functional assay like inhibition of cAMP or [^35^S] GTPγS assay would be more appropriate for this purpose. Was the BRET assay performed in the presence of sodium ion?

We thank the reviewer for this helpful comment. We find that our data are consistent with a previously published study, which like us, specifically investigated the relative intrinsic inverse agonism of various antagonists at the D_2_R, but in this case using a [^35^S]GTPγS assay (Roberts and Strange, 2005), which we have now described and cited in the subsection “Functional consequences of distinct antagonist-bound inactive conformations”. This study also found that (+)-butaclamol, risperidone, and spiperone displayed greater inverse agonism than raclopride and (-)-sulpiride. This illustrates, that our findings using the BRET assay measuring G protein activation is consistent with studies using another approach, [^35^S]GTPγS assay, to measure receptor activation. It should be noted, however, that the studies of Roberts and Strange were performed in the absence of Na^+^ in order to obtain a large signal window to observe inverse agonism whereas the BRET assay was performed in the presence of sodium ion as mentioned in the subsection “Bioluminescence resonance energy transfer (BRET) assay”. Taken together these data suggest that occupation of the Ile^3.40^ sub-pocket confers both Na^+^ insensitivity for binding and inverse agonism, such that Na^+^ sensitivity is negatively related with the extent of inverse agonism for the tested ligands. However, other structural elements likely also contribute to the extent of inverse agonism. We have now clarified this inference in our Results and Discussion.

We are sorry to have missed the important reference about the inverse agonism of dopamine receptors by Zhang et al., and have now cited it in the first paragraph of the Discussion. However, we could not find specific discrepancy between our results and previous work. Nevertheless, we have now limited our conclusion specifically to the ligands tested both in our study and in Roberts and Strange.

2) The MD findings related to EL1 and EL2 lacks adequate support from biochemical experiments. Testing a few mutations related to the EL1 and EL1 findings would strengthen this manuscript.

Following this suggestion, we have experimentally mutated Trp100 in EL1, Ile184 in EL2, and Leu94^2.64^, and evaluated the effects of the W100A, I184A, and L94A mutations on the binding affinities of eticlopride, risperidone, and spiperone. Our findings support our conclusions that the close interaction between Ile184 and Trp100 revealed by the D_2_R/risperidone crystal structure is not necessary for the stabilization of the risperidone pose (see the new subsection “The Ile184^EL2.50^-Trp100^EL1.50^ interaction is not critical for risperidone binding”).

3) Four ligands that occupy the Ile^3.40^ sub-pocket are docked, but only one that does not occupy the pocket. Given that Nemonapride is also a non-selective and high affinity D_2_R antagonist, this ligand should be included in the MD study.

To further characterize the Na^+^-sensitive ligands, we have carried out new molecular dynamics simulations of D_2_R in complex with (-)-sulpiride, which was experimentally tested in this study, and as the reviewer suggested, the D_2_R/nemonapride complex.

As shown in the updated Figure 2—figure supplement 1, similar to the conclusion reached in our previous study (Michino et al., 2015), the (-)-sulpiride pose is stable in the presence of the bound Na^+^ but not in its absence.

To the extent of our simulations within two months with the available resources, it appears that the accommodation of nemonapride in the D_2_R binding pocket likely requires significant rearrangement of Phe^3.28^, along with coordinated movements of the EL1 loop. It is premature to report our current results. Indeed, as indicated in the D_4_R crystal structure paper (Wang et al., 2017), “In DRD4, nemonapride’s unsubstituted benzyl group is wedged between F91^2.61^ and L111^3.28^, indicating that its binding to DRD3 likely requires structural rearrangements to avoid clashes with V86^2.61^ and F106^3.28^” – this is also very likely the situation in D_2_R in which the residues at 2.61 and 3.28 are identical to D_3_R.

In addition, we would like to point out that in Supplementary Figure 9 of the aforementioned D_4_R crystal structure paper, similar to our results of other Na^+^-sensitive ligands in D_2_R, the authors in that paper indicated that in D_4_R, “simulations also indicated that nemonapride’s binding pose is more stable in the presence of sodium”, which we have now cited in the subsection “Occupation of the Ile^3.40^ sub-pocket confers insensitivity to Na^+^ in antagonist binding”.

4) With regards to the different conformation identified in EL2, could the author comment on which conformation identified will likely be useful for virtual screening in drug discovery?

Our findings in this study prompted us to propose that limiting a virtual screening to one conformation will miss relevant ligands, and we have made this clearer in the revised manuscript. Indeed, the strategy of ensemble docking, in which each ligand is docked to a set of receptor conformers, has been adapted in recent virtual screening efforts.

To characterize the conformational ensemble sampled by D_2_R in complexes with ligands bearing different scaffolds, we clustered conformations of the orthosteric binding site (OBS) sampled in our representative D_2_R/eticlopride and D_2_R/risperidone trajectories in which EL2 transitioned from the helical to the unwound conformations. See the subsection “The clustering of the binding site conformations” and Table 4.

Our results indicate that i) as expected, the OBS conformations in these two complexes are significantly different and can be easily differentiated in our clustering; and ii)

the helical versus extended EL2 conformations are not closely associated with the OBS conformations.

Thus, while the centroid frames from each cluster in this clustering can form an ensemble for future virtual screening for the primary scaffold occupying the OBS, in order to discover novel prolonged ligands that protrude out of the OBS to interact with residues in EL2 and EL1, additional frames that cover both helical and extended EL2 conformations from each cluster will have be used to screen for the optimal extensions of the primary scaffold.

5) Subsection “Both the EL2 conformation and ligand scaffold affect the EL1 conformation”, second paragraph. Interestingly, in this position, the conformation of Trp^EL1.50^ can be stabilized by the disulfide bond of EL2 (Ioerger et al., 1999) (as shown in Video 2) or by interaction with the N terminus, which was truncated in the receptor construct used in the determination of the crystal structure. – Is there any evidence for the interaction with the N-terminus?

We do not have experimental evidence showing that Trp^EL1.50^ interacts with the N-terminus but as we reviewed recently in Verma et al., PLoS Comput Biol 2018, functional roles of the N-terminus have been documented in a few closely related D_2_R homologs.